# Virus adaptation to heparan sulfate comes with capsid stability tradeoff

Han Kang Tee[1]*, Simon Crouzet[2], Arunima Muliyil[1], Gregory Mathez[1], Valeria Cagno[1], Matteo Dal Peraro[2], Aleksandar Antanasijevic[3†], Sophie Clément[1†], Caroline Tapparel[1]*†

[1]Department of Microbiology and Molecular Medicine, University of Geneva, Geneva, Switzerland; [2]Interschool Institute of Bioengineering (SV), School of Life Sciences, École Polytechnique Fédérale de Lausanne (EPFL), Lausanne, Switzerland; [3]Global Health Institute, School of Life Sciences, École Polytechnique Fédérale de Lausanne (EPFL), Lausanne, Switzerland

*For correspondence:
han.tee@unige.ch (HKangT);
caroline.tapparel@unige.ch (CT)

†These authors contributed equally to this work

**Abstract** Because of high mutation rates, viruses constantly adapt to new environments. When propagated in cell lines, certain viruses acquire positively charged amino acids on their surface proteins, enabling them to utilize negatively charged heparan sulfate (HS) as an attachment receptor. In this study, we used enterovirus A71 (EV-A71) as the model and demonstrated that, unlike the parental MP4 variant, the cell-adapted strong HS-binder MP4-97R/167 G does not require acidification for uncoating and releases its genome in the neutral or weakly acidic environment of early endosomes. We experimentally confirmed that this pH-independent entry is not associated with the use of HS as an attachment receptor but rather with compromised capsid stability. We then extended these findings to another HS-dependent strain. In summary, our data indicate that the acquisition of capsid mutations conferring affinity for HS comes together with decreased capsid stability and allows EV-A71 to enter the cell via a pH-independent pathway. This pH-independent entry mechanism boosts viral replication in cell lines but may prove deleterious *in vivo*, especially for enteric viruses crossing the acidic gastric environment before reaching their primary replication site, the intestine. Our study thus provides new insight into the mechanisms underlying the *in vivo* attenuation of HS-binding EV-A71 strains. Not only are these viruses hindered in tissues rich in HS due to viral trapping, as generally accepted, but our research reveals that their diminished capsid stability further contributes to attenuation *in vivo*. This underscores the complex relationship between HS-binding, capsid stability, and viral fitness, where increased replication in cell lines coincides with attenuation in harsh *in vivo* environments like the gastrointestinal tract.

## Editor's evaluation

This is important work, correlating capsid stability with mutations that promote heparan sulfate binding. The data are solid and support the claims.

## Introduction

HS are linear, negatively charged polysaccharides connected to various cell surfaces and extracellular matrix proteins. Expressed on a wide range of cells, they play a pivotal role in various biological processes, and many viruses exploit them to attach and concentrate onto cell surfaces before binding to the main entry receptor (*Cagno et al., 2019*). Despite a substantial body of literature on the subject, the actual implication of HS binding on viral infections remains a topic of debate (*Cagno et al., 2019*).

Enterovirus A71 (EV-A71) is an excellent example of the ongoing controversy regarding the impact of HS receptor utilization in viral pathogenesis. This virus is a member of the *Picornaviridae* family and the most neurotropic EV after poliovirus. It causes significant hand, foot, and mouth disease outbreaks, particularly in Asia-Pacific countries, and is associated with severe neurological complications, notably in small children and immunosuppressed patients (*Tee et al., 2021*). The virus uses human scavenger receptor class B member 2 (SCARB2) as the main entry receptor for uncoating (*Kobayashi and Koike, 2020*; *Yamayoshi et al., 2013*). Since SCARB2 is mostly localized on endosomal and lysosomal membranes and sparsely on plasma membrane (*Kobayashi and Koike, 2020*; *Kuronita et al., 2002*), it seems to play only a minor role in EV-A71 cell attachment (*Nishimura et al., 2024*; *Guo et al., 2022*). Consistently, numerous other EV-A71 attachment receptors have been described in the literature, including HS (*Kobayashi and Koike, 2020*; *Dang et al., 2014*). When propagated in cell culture, EV-A71 rapidly acquires adaptive mutations (i.e. patches of positively charged amino acids on the viral capsid) that allow them to bind HS, sometimes with high avidity. These strong HS-dependent variants grow efficiently in cell culture but show attenuated virulence in animal models, such as mice and cynomolgus monkeys (*Chua et al., 2008*; *Fujii et al., 2018*; *Kobayashi et al., 2018*; *Tee et al., 2019*). Analysis of the differential expression of SCARB2 and HS in tissues from monkey or transgenic mice revealed little overlap. Strong HS expression was detected in sinusoidal endothelial cells and vascular endothelia, where SCARB2 was not detected (*Fujii et al., 2018*; *Kobayashi et al., 2018*). Similarly, HS expression in the brain was mainly found in vascular endothelia but SCARB2 expression was found predominantly in neuronal cells. The authors of these studies concluded that binding to HS on endothelial cells in the absence of SCARB2 leads to viral trapping, abortive infection, and attenuation (*Fujii et al., 2018*; *Kobayashi et al., 2018*). Similar observations were shown for other viruses, including Murray Valley encephalitis (*Lee and Lobigs, 2002*), Japanese encephalitis (*Lee and Lobigs, 2002*), Sindbis (*Byrnes and Griffin, 2000*), Theiler's murine encephalomyelitis (*Reddi and Lipton, 2002*), tick-borne encephalitis (*Mandl et al., 2001*), West Nile (*Lee et al., 2004*) and dengue (*Chen et al., 1997*). However, epidemiological surveillance of human EV-A71 infections (*Chang et al., 2012*; *Liu et al., 2014*; *Cheng et al., 2014*) and experimental evidence from 2D human fetal intestinal models (*Aknouch et al., 2023*), human airway organoids (*van der Sanden et al., 2018*), and air-liquid interface cultures (*Tseligka et al., 2018*) suggest that HS binding may enhance viral replication and virulence in humans. In addition, recent research has shown that EV-A71 can be released and transmitted via cellular extrusions (*Moshiri et al., 2023*) or exosomes (*Huang et al., 2020*), potentially preventing viral trapping of HS-binding strains in the circulation. Further studies are required to evaluate the true impact of HS-binding mutations on the spread and virulence of EV-A71 in both animal models and humans.

We previously isolated cell-adapted EV-A71 mutants with strong affinity for HS which emerged upon passaging of intermediate HS binders derived from both patient and mouse-adapted MP4 strains in cell culture (*Cordey et al., 2012*; *Weng et al., 2023*). The mutants presented two amino acid changes in the VP1 capsid protein: VP1-L97R mutation in the VP1 BC loop, shown to confer intermediate affinity for HS together with a secondary mutation, VP1-E167G, located in the VP1 EF loop, which significantly strengthened HS binding with reduction of negative charges (*Tseligka et al., 2018*; *Weng et al., 2023*). As previously observed for strong HS-binding variants, we showed that, in contrast to the original mouse-adapted MP4 strain which exhibited virulence in mice, this cell-adapted MP4-97R/167G double mutant was completely attenuated in mice (*Weng et al., 2023*). In the current study, we used MP4 and MP4-97R/167G mutant as representatives of respectively, weak and strong HS-binders, slow and fast-growing in cell lines and virulent and avirulent in mouse models (as documented previously *Tseligka et al., 2018*; *Weng et al., 2023*), to elucidate the consequence of virus adaptation towards HS binding on the viral growth cycle. We demonstrated that these mutations not only increase binding to HS, but also reduce capsid stability, leading to improved uncoating and faster cell internalization in a HS-independent manner. Of note, another strong HS-binder harboring VP1-E145Q substitution also showed decreased capsid stability compared to the wildtype HS-independent variant. These data provide another possible explanation for the *in vivo* attenuation of strong HS-binders which may originate from viral trapping but also from decreased capsid stability which would be detrimental to the virus in challenging environments such as the gastrointestinal tract.

## Results

### Lysosomotropic drugs reduce the infectivity of the HS-independent MP4 but enhance the infectivity of the strong HS-binder MP4-97R/167G

First, we sought to assess whether viruses displaying different dependence on HS exploit different growth cycle pathways. We compared the effect of lysosomotropic drugs, namely hydroxychloroquine (HCQ) and bafilomycin A1 (BAF-A1) on MP4 and MP4-97R/167G double mutant. As presented in *Figure 1A*, Vero cells were pre-treated for 1 hr with each drug before infection. These drugs showed no cytotoxic effect at the concentrations used in the assay (*Figure 1—figure supplement 1A*). Inoculation was then performed for 1 hr in the presence of the drug and inoculum was removed and replaced with fresh drug-free media. To confirm the inhibition of endosomal acidification by the drugs, the presence or absence of acidic lysosomes was assessed by immunostaining of lysosomal-associated membrane protein 1 (LAMP-1) and by staining with LysoTracker, a dye specific for acidic compartments. The signal intensity of LysoTracker decreased drastically following treatment with the drugs, confirming the inhibition of endosomal acidification (*Figure 1—figure supplement 1B and C*). The effect of the drugs on viral replication was then compared for the two variants (*Figure 1B* and *Figure 1—figure supplement 1D*). MP4 infectivity was significantly reduced by both drugs in a dose-dependent manner, while MP4-97R/167G infectivity was in contrast enhanced. These results were confirmed by viral load quantification with real-time RT-PCR (*Figure 1—figure supplement 1E*). Similar results were obtained in RD cells (*Figure 1—figure supplement 1F and G*), indicating that drug effects are not cell type-dependent. The different sensitivity of the two variants to acidification inhibitors was more pronounced with HCQ, so we performed a detailed examination of the mechanism of action with this drug.

To determine whether the effect of HCQ was related to the usage of HS as an attachment receptor, we repeated the virus inhibitory assay using cells depleted of HS by either heparinase digestion or treatment with sodium chlorate (*Figure 1C*). The distinct sensitivity to HCQ was reproduced regardless of the presence or absence of HS on the cell surface. Of note, we confirmed that as for the human strain (*Tseligka et al., 2018*), the HS-independent and HS-dependent MP4- derivatives both need SCARB2 to infect cells and cannot replicate in SCARB2 CRISPR-Cas9 knock-out cells (*Figure 1—figure supplement 2*). Altogether, these data indicate that the capsid mutations change the sensitivity to HCQ, independent of the attachment receptor used.

### MP4 enters via SCARB2-mediated and pH-dependent endocytosis, while MP4-97R/167G utilizes an alternative SCARB2-dependent pathway

We showed that the MP4 virus uses SCARB2 as an entry receptor and that it is inhibited by HCQ. This strongly suggests that MP4 uses SCARB2-mediated pH-dependent endocytosis for entry as demonstrated for many other EV-A71 variants (*Dang et al., 2014*; *Chen et al., 2012*). We then evaluated whether HCQ could inhibit viral binding. As anticipated, due to its affinity for HS, MP4-97R/167G showed higher cell binding compared to MP4. However, HCQ and no effect on this binding (*Figure 2A*). We next conducted a single-cycle infection assay and observed that the differential effect of HCQ on MP4 and MP4-97R/167G became prominent from 4 hpi onward (*Figure 2B*). To further dissect which step of the viral growth cycle was differentially affected by the treatment, we performed a time-of-addition assay. As shown in *Figure 2C*, HCQ significantly lost its effect when administered later than 1 hpi, confirming that the effect occurs during the early phase of the viral cycle. To more specifically assess whether the drug affects viral entry, we transfected *in vitro* transcribed genomic RNA containing the nanoluciferase (Nluc) gene as the reporter (*Figure 2D and E*). RNA transfection allows to bypass receptor-mediated entry. In these conditions, no difference was observed for both variants, whether HCQ was present or not (*Figure 2E*). This observation also indicates that the drug does not impact genome replication. In contrast, infection with infectious Nluc reporter virus (*Figure 2D and F*) reproduced the differential HCQ inhibition as observed in the original non-modified viruses (*Figure 1D*). Taken together, these data indicate that MP4 enters via pH-dependent endocytosis, while MP4-97R/167G entry pathway is independent on endosomal acidification.

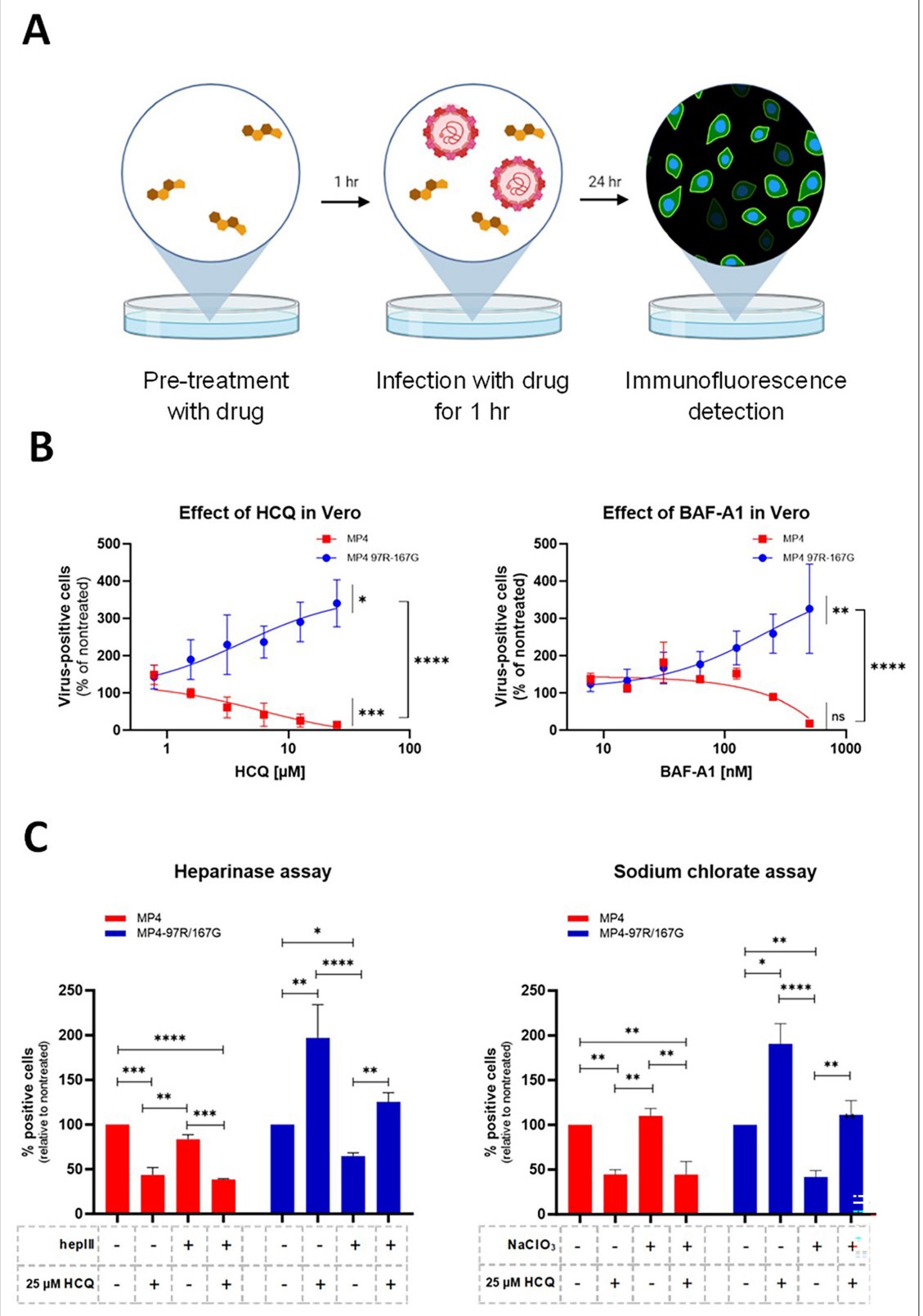

**Figure 1.** Lysosomotropic drugs inhibit infection by MP4 but not by MP4-97R/167G. (**A**) Schematic illustration of the virus inhibitory assay workflow. Cells were pre-treated with lysosomotropic drugs and infected (MOI 0.1) in presence of the drug. After inoculum removal, infected cells were cultured in drug-free media and infected cells were stained by immunofluorescence (IF) with anti-VP2 Ab. Figure 1A was created with BioRender.com. (**B**) Hydroxychloquine (HCQ) and Bafilomycin A1 (BAF-A1) dose response assay in infected Vero cells. (**C**) HCQ effect in Vero cells pre-treated or not

*Figure 1 continued on next page*

*Figure 1 continued*

with heparinase III (hepIII) or sodium chlorate (NaClO₃) as in A. Mean and S.E.M of biological triplicates are shown. Results are shown as % of virus-positive cells relative to nontreated control. In B, statistical significance (one-way ANOVA) between treated and untreated virus or between treated MP4 and MP4-97R/167G was calculated based on the area under curve (AUC). In C, statistical significance (two-way ANOVA) was calculated for each virus between each condition. *p<0.05, **p<0.01, ***p<0.001, ****p<0.0001.

The online version of this article includes the following source data and figure supplement(s) for figure 1:

**Source data 1.** Related to *Figure 1B*.

**Source data 2.** Related to *Figure 1C*.

**Figure supplement 1.** Toxicity and efficacy of lysosomotropic drugs.

**Figure supplement 1—source data 1.** Related to *Figure 1—figure supplement 1A*.

**Figure supplement 1—source data 2.** Related to *Figure 1—figure supplement 1C*.

**Figure supplement 1—source data 3.** Related to *Figure 1—figure supplement 1E*.

**Figure supplement 1—source data 4.** Related to *Figure 1—figure supplement 1F*.

**Figure supplement 1—source data 5.** Related to *Figure 1—figure supplement 1G*.

**Figure supplement 2.** Virus infection in SCARB2-KO cells.

**Figure supplement 2—source data 1.** Related to *Figure 1—figure supplement 2*.

## HCQ differentially impacts MP4 and MP4-97R/167G uncoating

To further dissect the mechanism of action of HCQ on the entry of each variant, we next examined the effect of the drug on viral uncoating. We generated virus stocks labeled with neutral red and performed a neutral red uncoating assay, as previously described (*Brandenburg et al., 2007*). Briefly, viral stocks were produced in the dark in the presence of neutral red to allow co-encapsidation of the viral genome and the dye within the viral particles. Photoactivation of neutral red causes the dye to cross-link viral genomes to the capsid and block viral uncoating (*Brandenburg et al., 2007*). Thus, upon infection with neutral red-labeled viruses, light exposure only inactivates viruses that have not yet completed uncoating, while viral genomes already released in the cytoplasm remain unaffected. This technique precisely determines the timepoint of viral uncoating. Cells were pre-treated with HCQ and then incubated with neutral red-labeled viruses for 1 hr at 37 °C for infection. Light inactivation was performed at selected timepoints post-infection, and infected cells were quantified 24 hr later by immunostaining (*Figure 3A*). In the absence of HCQ, most viruses had undergone uncoating between 2 and 4 hpi for both variants (30–80% of uncoated viruses for MP4 and 45–90% for MP4-97R/167G, respectively) (*Figure 3B*). In the presence of HCQ, MP4 uncoating was greatly inhibited, even when photoactivation was performed at 4 hpi (*Figure 3B*, **left panel**). In contrast, the uncoating rate was not inhibited in the presence of HCQ for MP4-97R/167 G (*Figure 3B*, **right panel**), and the final viral yield was even increased as already observed in *Figure 1B and C*.

To further validate these results, we then combined fluorescent *in situ* hybridization (FISH) of viral genomic RNA and immunofluorescence staining of viral capsid at early timepoints. Full particles are characterized by colocalization of virus genomic RNA (vRNA) and viral capsid (as shown in 1 hpi at 4 °C), while the colocalization is lost following the uncoating process (*Figure 3C*). Quantification of vRNA and capsid colocalization highlighted no significant difference between the two variants at 2 hpi in presence or absence of HCQ (*Figure 3D*). However, at 4 hpi (prior to the initiation of replication, see *Figure 3B*), MP4 uncoating appeared to be inhibited by HCQ, as highlighted by a decrease of empty capsids and an increase of capsid/RNA colocalization in presence of the drug (*Figure 3D and E*). An opposite effect was observed for MP4-97R/167G, with a reduced capsid/RNA colocalization in the presence of HCQ, indicating that more viruses had undergone uncoating in the presence of HCQ at this time point. Altogether these data indicate that MP4-97R/167G can uncoat at neutral pH and that acidification is instead increasing its replication capacity, while MP4 needs acidic pH to uncoat.

## MP4 relies on late endosomes for uncoating, whereas MP4-97R/167G undergoes uncoating in early endosomes

HCQ is known to inhibit endosomal acidification by accumulating in endosomes in a protonated form. This accumulation leads to endosomal swelling and inhibition of fusion between endosomes and lyso-somes within cells, as previously described (*Baradaran Eftekhari et al., 2020*; *Mauthe et al., 2018*)

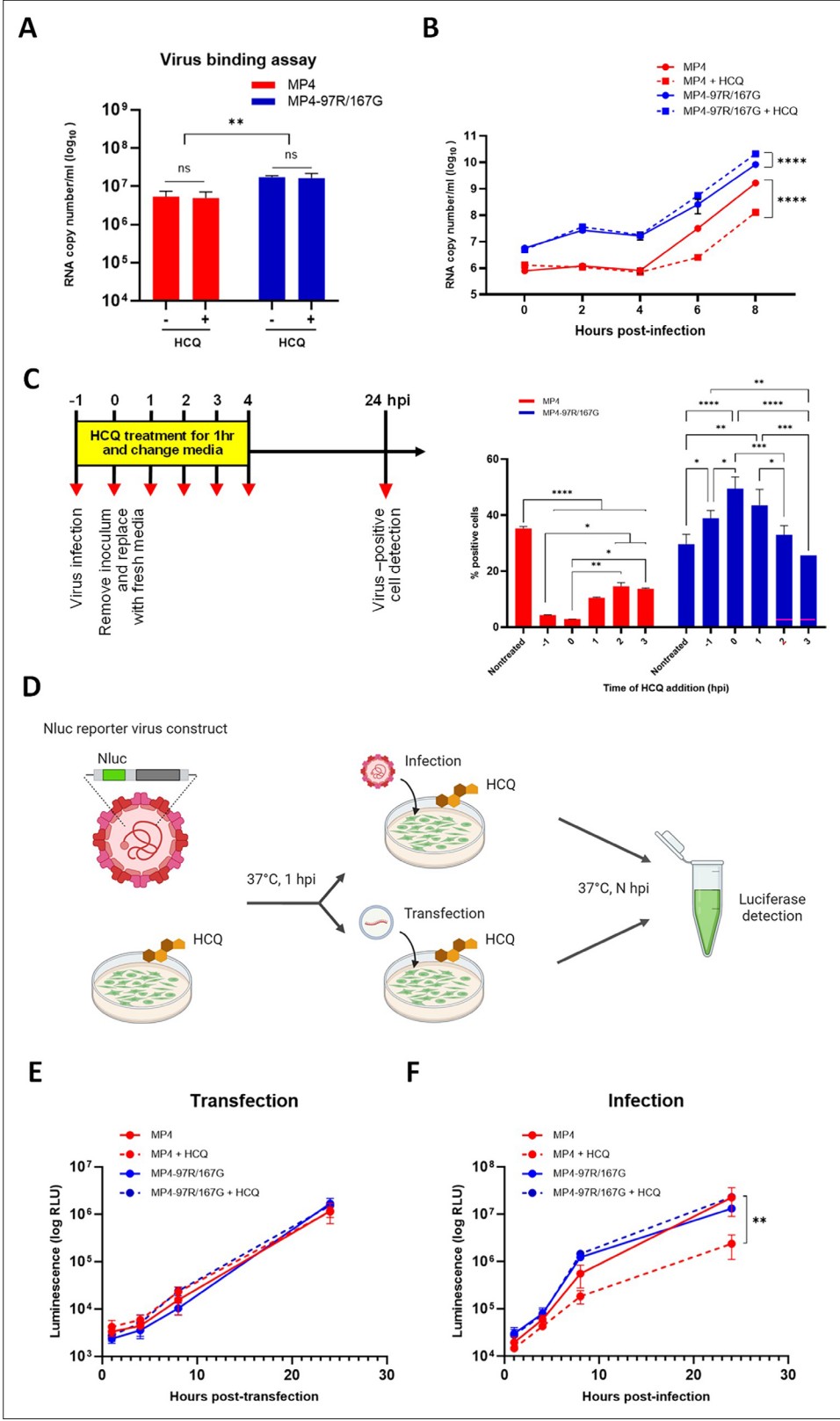

**Figure 2.** Hydroxychloroquine (HCQ) targets viral entry. (**A**) Virus binding assay in Vero cells in presence of 25 µM HCQ. (**B**) Single-cycle replication kinetic in nontreated and HCQ-treated Vero cells. At each timepoint, cell lysates were collected, and viral RNA copy numbers were quantitated using RT-qPCR (**C**) Time-of-addition assay in Vero cells treated with HCQ starting at different timepoints. Infected cells (MOI 0.1) were quantitated

*Figure 2 continued on next page*

*Figure 2 continued*

24 hpi by immunofluorescence (IF). (**D**) Schematic diagram of Vero cells pre-treated with HCQ and subsequently subjected to transfection of *in vitro* RNA transcript or infection with EV-A71 nanoluciferase (Nluc) reporter viruses. At the indicated timepoints, cell supernatants were collected, and luciferase activity was measured. Figure 2D was created with BioRender.com.(**E & F**) Results are expressed in % relative light unit (RLU) of treated versus nontreated virus at indicated timepoints. The mean and S.E.M from biological triplicates are shown. Statistical significance was calculated using two-way ANOVA, comparing treated and untreated control. *p<0.05, **p<0.01, ***p<0.001, ****p<0.0001.

The online version of this article includes the following source data for figure 2:

**Source data 1.** Related to *Figure 2A*.

**Source data 2.** Related to *Figure 2B*.

**Source data 3.** Related to *Figure 2C*.

**Source data 4.** Related to *Figure 2E*.

**Source data 5.** Related to *Figure 2F*.

---

and as shown in *Figure 4A*. We thus hypothesized that the two variants could exploit different entry routes, which could explain the different sensitivity to HCQ. We showed that MP4 needs acidic pH to uncoat and is thus expected to release its RNA in late endosomes/lysosomes. In contrast, MP4-97R/167G can uncoat in the absence of pH acidification and accordingly in a non-acidic environment. To test this hypothesis, we infected Vero cells transiently expressing a variant of small GTPase Rab5a, a protein involved in the maturation of early endosomes (EE) into late endosomes (LE). This Rab5a-Q79L mutant is constitutively active (CA) and blocks LE maturation (*Figure 4B*). Viral capsids of MP4 and MP4-97R/167G were observed within EE in both Rab5a WT and CA-expressing cells at 0.5 hpi (*Figure 4—figure supplement 1A*) and 2 hpi (*Figure 4C*). However, at 7 hpi, the percentage of cells stained for double-stranded RNA (dsRNA), a marker of virus replication, was significantly reduced for MP4 in Rab5a CA-expressing cells but not for MP4-97R/167G (*Figure 4D* and *Figure 4—figure supplement 1B*). This indicates that MP4-97R/167G genomes were successfully released in the cytoplasm to undergo translation and replication, even in the absence of EE fusion to LE. Conversely, a transition from EE to LE with a gradual pH decrease is necessary for MP4 to release its genome.

## VP1-L97R/E167G substitutions confer affinity for HS but decrease capsid stability

Our data highlighted that both MP4 and MP4-97R/167G enter via a SCARB2-dependent pathway, and localize in early endosomes at early times post-infection but exhibit distinct sensitivities to HCQ, a feature independent of their differential use of HS as an attachment receptor. We therefore speculated that the varying pH-dependency may be attributed to differences in virion stability. We analyzed the impact of VP1-97R and VP1-167G mutations on their respective local environments at pH 5 and pH 7 with the Adaptive Poisson-Boltzmann Solver (APBS) (*Bank and Holst, 2003*; *Jurrus et al., 2018*). As displayed in *Figure 5—figure supplement 1*, both L97R and E167G mutations are inducing electrostatic changes at the surface of the capsid within the region of interest and the changes are particularly significant at pH 5. Furthermore, analysis with DynaMut server (*Rodrigues et al., 2018*) reveals that the two mutations affect interaction networks (*Figure 5—figure supplement 2*). The VP1-L97R mutation is predicted to reduce hydrophobic interactions between amino acid 97 and its neighbors VP1-245Y and VP1-246P while the VP1-E167G mutation causes a loss in hydrogen bonding capacity to VP1-165S and reduces the net negative charge. This is consistent with analyses of vibrational entropy change (*Figure 5—figure supplement 3*), indicating that the presence of two mutations results in enhanced local dynamics, which has previously been correlated with reduced capsid stability (*Panja et al., 2020*; *Chand et al., 2020*). Consistently, the predictions of Gibbs free energy change (ΔΔG) induced by these mutations further support that both mutations induce destabilization of the capsid structure, regardless of pH and temperature (*Figure 5—figure supplement 1*).

These computational predictions led us to speculate that the MP4-97R/167G mutant may feature a lower stability and may be able to bypass the need for acidic pH for uncoating. To experimentally test this hypothesis, we subjected these variants to neutral and acidic conditions and assessed their virion structure using negative staining electron microscopy (nsEM) (*Figure 5A and B*). At acidic pH,

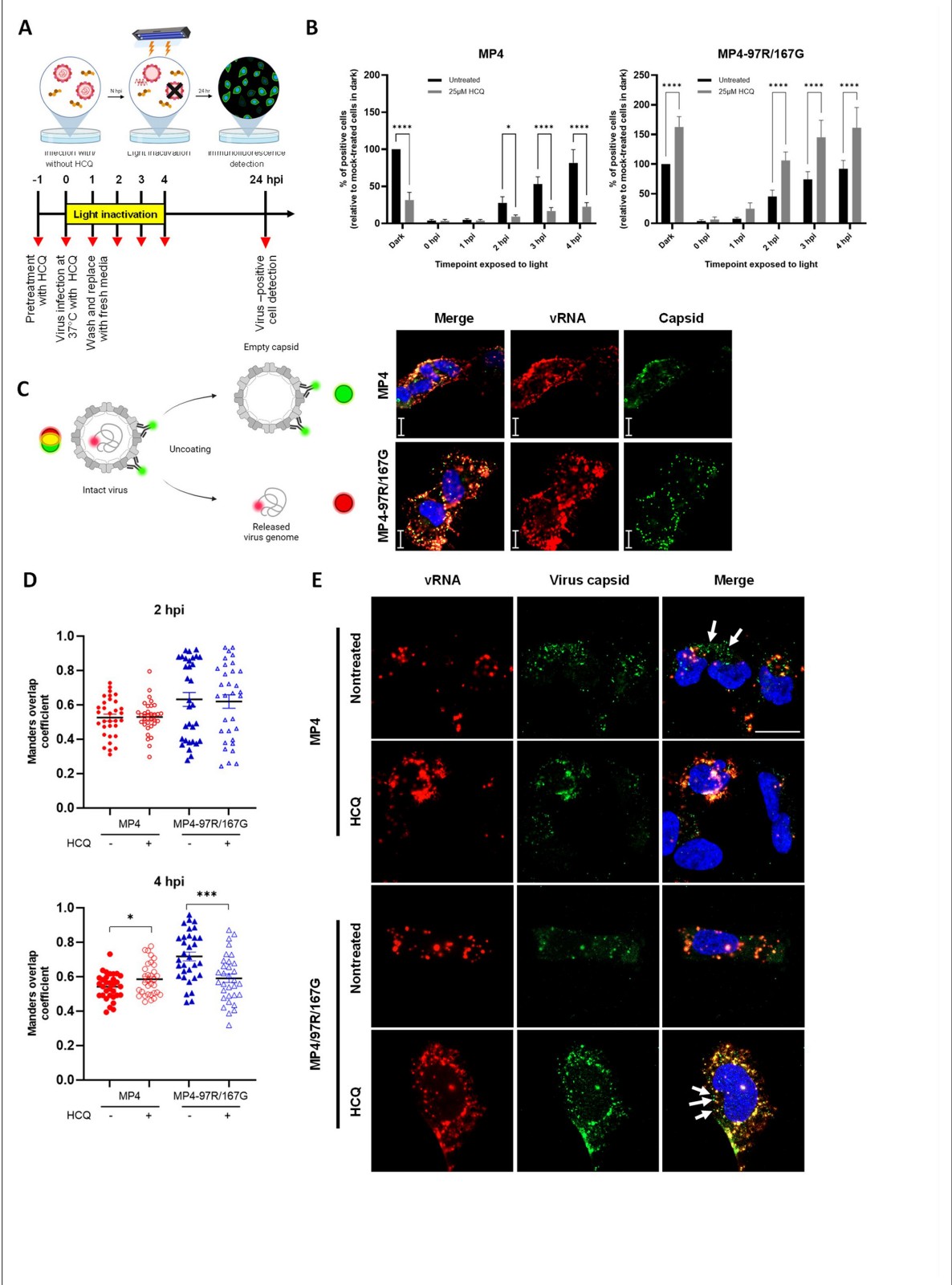

**Figure 3.** Hydroxychloroquine (HCQ) delays the uncoating of MP4. (**A**) Schematic illustration of the neutral red assay workflow. Vero cells were pre-treated with or without HCQ for 1 hr. Neutral red-labeled viruses were allowed for cell infection at 37 °C for 1 hr (MOI 0.1). The inoculum was then removed and replaced with fresh media. Infected cells were exposed to light for 30 min at different timepoints and further incubated up to 24 hpi for immunofluorescence (IF) staining. Figure 3A was created with BioRender.com. (**B**) Effect of light inactivation on replication of neutral red-labeled

*Figure 3 continued on next page*

*Figure 3 continued*

MP4 (left panel) or MP4-97R/167G (right panel). Results are plotted as % of virus-positive cells relative to non-treated dark control. Mean and S.E.M of biological triplicates are shown. Statistical significances (two-way ANOVA) were calculated between treated and nontreated conditions. (**C**) Schematic illustration of virus uncoating monitored with the combinational use of RNA-FISH to detect EV-A71 RNA (red) and IF with anti-VP2 Ab to detect the viral capsid (green). Co-staining highlights intact viruses in yellow while empty capsids and free RNA are in green and red, respectively. (C, right panel) Representative images (scale, 20 μm) of MP4 and MP4-97R/167G binding after 1 hr at 4 °C with virus genomic RNA (vRNA) (red) and capsids (green). Figure 3C was created with BioRender.com. (**D**) Co-localization of capsid and vRNA in individual cells at 2 hpi and 4 hpi analyzed using Mander's overlap coefficient (n=32 individual cells from two independent experiments). Statistical comparison (unpaired t-test) of untreated and treated groups. *p<0.05, **p<0.01, ***p<0.001, ****p<0.0001. (**E**) Representative images of the 4 hr time point. Arrows: empty capsid.

The online version of this article includes the following source data for figure 3:

**Source data 1.** Related to *Figure 3B*.

**Source data 2.** Related to *Figure 3D*.

there was no notable alteration in the capsid morphology of MP4, which maintained a stable particle diameter of ~31–33 nm across both pH conditions. On the other hand, for MP4-97R/167G, both 2D images and 3D reconstructions highlighted a loss of density at the center of the viral particles, as well as an expansion in size for a subset of particles (diameter ranging from 31 to 41 nm at pH5 versus 31 to 33 nm at pH7), indicating partial virus uncoating. We then performed a temperature sensitivity assay by heating viruses at different temperatures for 1 hr before inoculation on cells. Quantification of infected cells at 24 hpi further confirmed that MP4 capsid is more resistant to higher temperature as 80% of the MP4 population survived a 50 °C thermal stress compared to only 50% for MP4-97R/167G (*Figure 5C*).

As MP4-97R/167G is less stable, we hypothesized that binding to SCARB2 may be sufficient to trigger its capsid opening. We conducted a competitive experiment and compared the infectivity of the two variants after incubation with soluble SCARB2 (sSCARB2) at neutral pH for 1 hr at 37 °C. We observed that MP4-97R/167G but not the MP4 variant lost infectivity upon pre-exposure to sSCARB2 (*Figure 5D*). We further tested the impact of the uncoating receptor binding by nsEM. MP4 and MP4-97R/167G were incubated with sSCARB2 at pH7.5 for 1 hr and imaged. Based on the inspection of EM images and 2D class-averages of MP4, the percentage of empty capsid particles did not change significantly before and after the incubation with sSCARB2 (<1% in both cases). However, the double mutant exhibited a ~ fivefold increase in empty capsid percentage after treatment with sSCARB2 (*Figure 5—figure supplement 4*), consistent with the functional data above. Altogether, our results confirmed that the MP4-97R/167G mutant is less stable and more sensitive to thermal and acidic stresses as well as receptor binding, which are sufficient triggers to initiate virus capsid disruption and subsequent viral uncoating.

## Resistance to HCQ and reduced capsid stability extend to other strong heparan sulfate-binding strains

To determine if our observations are applicable to human strains, we examined the sensitivity of a closely related clinical strain. This strain was isolated from the respiratory tract of an immunosuppressed patient with a disseminated EV-A71 infection (*Cordey et al., 2012*). Additionally, we tested a strong HS-binding derivative that harbors the same VP1-L97R and E167G mutations as our MP4 double mutant. Notably, this human clinical strain shares 98.3% amino acid similarity with the MP4 variant used in this study and exhibits similar HS-binding phenotypes (*Weng et al., 2023*). As shown in *Figure 6—figure supplement 1*, the original human strain was inhibited by HCQ, whereas the double mutant exhibited insensitivity to the drug. We next checked whether our findings could extend to other mutations conferring HS-binding ability. To this end, we used the human EV-A71 strain 41 (5865/SIN/000009, GenBank accession no. AF316321; subgenogroup B4), with or without a mutation at position VP1-145, a residue known to play a key role in modulating viral HS-binding capacity and *in vivo* virulence. The variant with VP1-145E is a weak HS binder and is virulent in mice while the cell-adapted VP1-145Q variant is a strong HS-binder and attenuated in mice (*Tee et al., 2019*). As shown in *Figure 6A*, HCQ enhanced the infectivity of the VP1-145Q variant but reduced the infectivity of the VP1-145E variant, aligning with our findings for the MP4-97R/167 G and its human-related strain (**Fig. S6**). We also investigated the effects of temperature and preincubation with sSCARB2, and confirmed the increased temperature sensitivity (*Figure 6B*) and sSCARB2 inhibition (*Figure 6C*) of

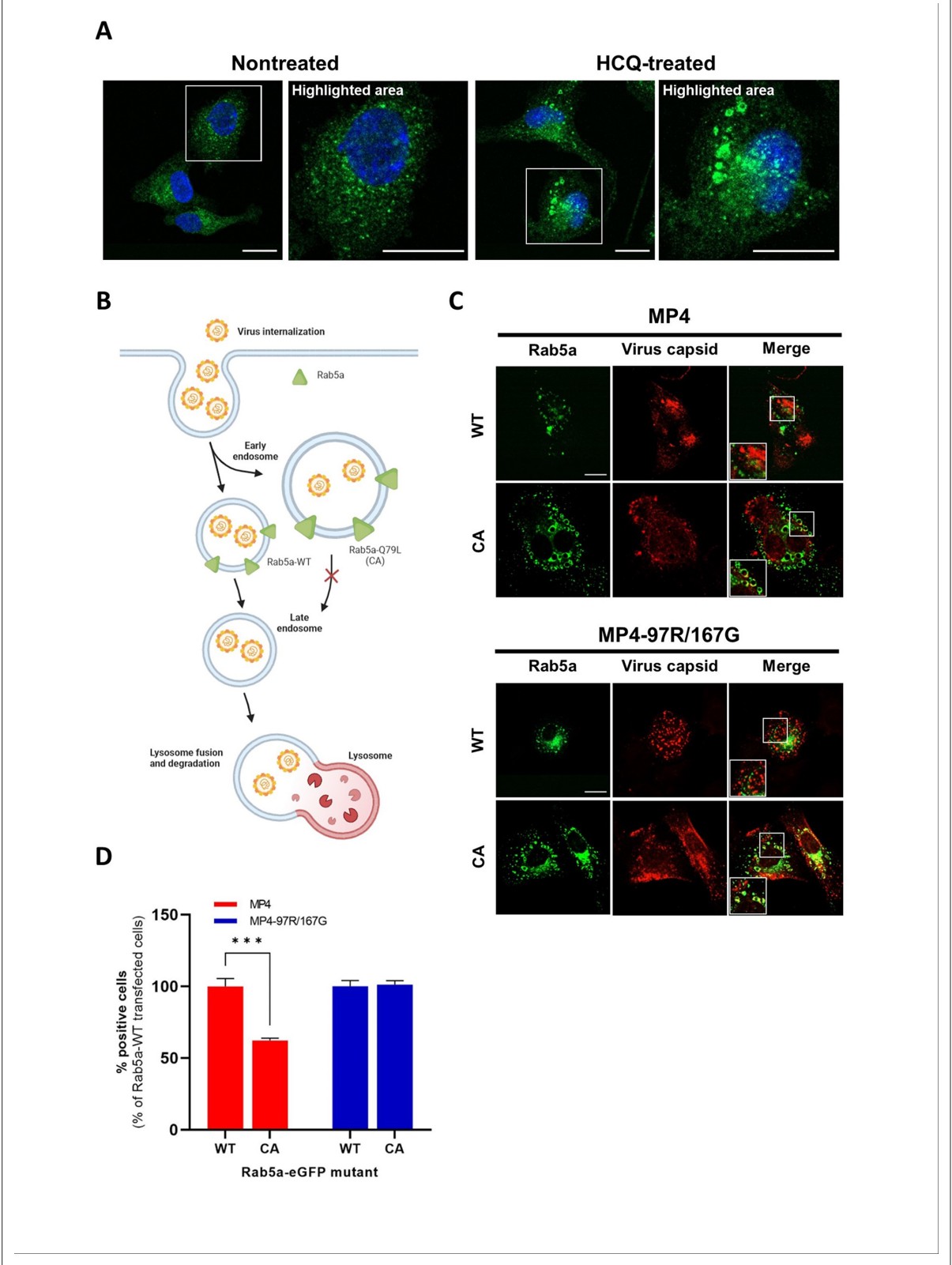

**Figure 4.** MP4-97R/167G uncoats from early endosomes. (**A**) Nontreated and hydroxychloroquine (HCQ)-treated Vero cells were stained with anti-EEA-1 antibody (green) to label early endosomes, and DAPI (blue) to label cell nuclei. (**B**) Schematic representation of endosomal route upon overexpression of Rab5a WT or constitutively active (CA) mutant. In C and D, Vero cells transiently expressing Rab5a-eGFP WT or CA were fluorescence-activated flow cytometry (FACS)-sorted and infected with the two viral variants. Infections were compared at different time post-infection. Figure 4B was created with

*Figure 4 continued on next page*

*Figure 4 continued*

BioRender.com. In (**C**) viral capsids (anti-VP2 Ab, in red) localize in early endosomes at 2 hpi in cells expressing Rab5a WT or CA. In (**D**) the proportion of cells containing replicating viruses (stained with the anti-dsRNA J2 Ab, see Fig.S3B for representative images) is calculated at 7 hpi. Results and statistical significance (two-way ANOVA) are expressed relative to cells with Rab5a WT. Mean and S.E.M from triplicates are shown. ***$p<0.001$. In B and C, white boxes are enlarged in the right panel. Scale bar: 20 μm.

The online version of this article includes the following source data and figure supplement(s) for figure 4:

**Source data 1.** Related to *Figure 4D*.

**Figure supplement 1.** Localization of viral capsids and double-stranded RNA at respectively 0.5 hpi and 7 hpi in fluorescence-activated flow cytometry (FACS)-sorted GFP-positive Vero cells transiently expressing Rab5a-eGFP WT or constitutively active (CA) and infected with the two viral variants.

the HS-binding VP1 145Q. In contrast, the VP1-145E variant's capsid proved to be much more stable. These experimental results are consistent with the free energy change prediction shown in *Figure 5—figure supplement 1*, further supporting the observation that the HS-binding phenotype is inversely correlated with virus capsid stability.

## Discussion

Acidic pH is an important trigger for viral uncoating and many enveloped or non-enveloped viruses, including influenza A virus (*White et al., 1982*; *Yoshimura et al., 1982*), human adenovirus (*Greber et al., 1993*), foot-and-mouth disease virus (*Vázquez-Calvo et al., 2012*), Semliki forest virus (*White et al., 1980*), enter the cell through pH-dependent endocytosis (*Ravindran and Tsai, 2016*). Similarly, for EV-A71, binding to SCARB2 and subsequent endosomal acidification are required for uncoating (*Yamayoshi et al., 2013*; *Dang et al., 2014*). In this study, we provide new insights on the impact of mutations in the VP1 capsid protein leading to strong HS affinity, on the uncoating process of EV-A71. We show that, unlike the mouse-adapted EV-A71 MP4 strain, the MP4-97R/167G-derived double mutant, which has a high affinity for HS, does not require acidification for uncoating and can release its genome under neutral or weakly acidic environment of early endosomes.

To demonstrate the importance of acidification on both MP4 and MP4-97R/167G variant uncoating, we used two lysosomotropic drugs, namely BAF-A1 and HCQ, that increase endosomal pH by distinct means. On one hand, BAF-A1 inhibits the vacuolar H+ATPase (V-ATPase), preventing the acidification process and thereby elevating the endosomal pH (*Yoshimori et al., 1991*; *Redmann et al., 2017*; *Wang et al., 2021*; *Yamamoto et al., 1998*; *Mauvezin and Neufeld, 2015*). On the other hand, HCQ, a less toxic derivative of the antimalarial drug chloroquine, acts as a weak base that can be protonated and trapped in the acidic environment of cellular organelles (*Pillat et al., 2020*; *Schrezenmeier and Dörner, 2020*; *Tian et al., 2021*). In addition to this effect, HCQ can impact other cellular pathways, such as autophagy, a cellular process that has been demonstrated to be induced by EV-A71 to create a favourable environment for its replication (*Xu et al., 2018*; *Huang et al., 2009*). However, we show here that the differential effects on MP4 and MP4-97R/167G occur during the uncoating process rather than in later stages of the cycle, such as virus genome replication (*Schrezenmeier and Dörner, 2020*). Our results thus underline that, despite their distinct modes of action, both HCQ and BAF-A1 influenced virus entry through their effect on endocytic compartments, as both compounds ultimately inhibit the reduction in endosomal pH levels. In addition, the fact that differential sensitivity to HCQ was retained even in cells devoid of HS at their surface by treatment with heparinase or sodium chlorate pointed out that this pH-independent mode of entry of MP4-97R/167G is not linked to the use of HS as an attachment receptor. Interestingly, our experiments using the Rab5a CA to block the transition of EE to acidic LE (*Bucci et al., 1992*; *Serio et al., 2011*), indicate that binding to SCARB2 is sufficient to trigger MP4-97R/167G genome release into the cytosol even in the near neutral pH of the EE (pH ~6.0–6.5), whereas MP4 requires the acidity of LE and/or lysosomes (pH ~5.0–5.5) to uncoat efficiently (*Mercer et al., 2010*). These observations led us to hypothesize that the two variants exhibit intrinsic differences in capsid stability. We conducted various tests to compare how each variant reacted to heat, sSCARB2, and low pH, and found that MP4-97R/167G was more sensitive to all these conditions. Particularly, we used nsEM to study the properties of viral particles and observed an expansion of MP4-97R/167G capsid following the exposure to pH 5. These data, plus the virus structural dynamics prediction and free energy change computation, all indicate that MP4-97R/167G presents reduced capsid stability compared to MP4.

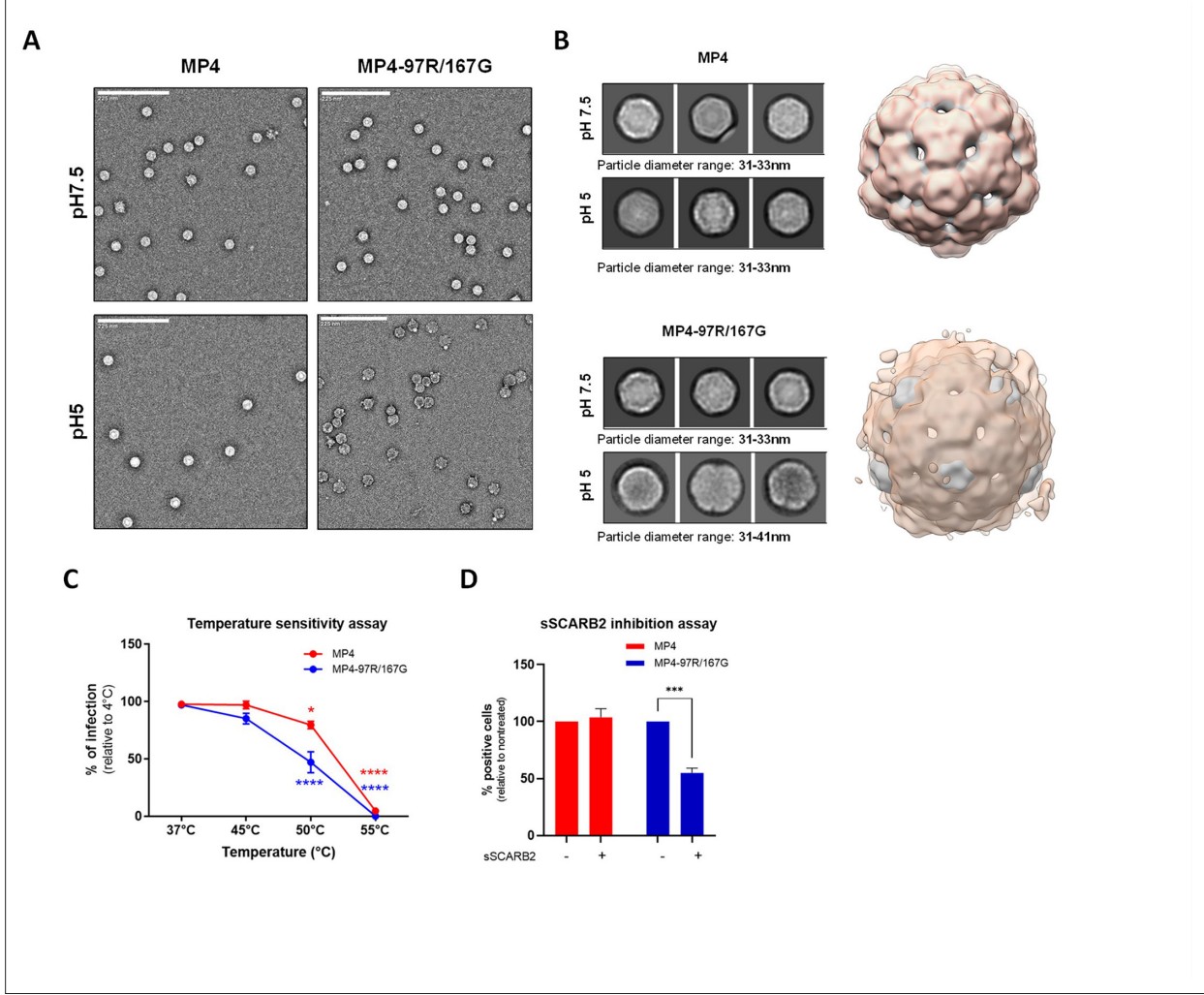

**Figure 5.** MP4 displayed stronger capsid stability and reduced sensitivity to acidification and high temperatures. (**A**) Negative staining electron microscopy (nsEM) analysis of MP4 and MP4-97R/167G incubated at pH 7 and pH 5. Representative raw micrographs are shown in each case. (**B**) Representative 2D class averages generated from datasets shown in panel A (box size = 54 nm; left) and the overlay of the corresponding 3D maps (right). Gray and orange shade indicates virus particle reconstructions at pH 7 and pH 5, respectively. (**C**) Temperature sensitivity assay. Infected Vero cells (MOI 0.5) were quantitated by immunostaining with an anti-VP2 Ab at 24 hpi after 1 hr incubation at increasing temperatures. Results are shown as % of virus-positive cells relative to 4 °C treated control. Error bars indicate mean and S.E.M from biological triplicates. (**D**) For sSCARB2 inhibition assay, viruses (MOI 0.5) were incubated 1 hr at 37 °C with 1 μg of soluble scavenger receptor class B member 2 (SCARB2) (sSCARB2) before infection of Vero cells. Infected Vero cells were quantitated by immunostaining with an anti-VP2 Ab at 24 hpi. Results are shown as % of virus-positive cells relative to nontreated controls. Statistically significance was calculated with two-way ANOVA. ***p<0.001, ****p<0.0001.

The online version of this article includes the following source data and figure supplement(s) for figure 5:

**Source data 1.** Related to *Figure 5C*.

**Source data 2.** Related to *Figure 5D*.

**Figure supplement 1.** Visual presentation of prediction of changes induced by the L97R/E167G mutations on electrostatic surface potential.

**Figure supplement 2.** Prediction of changes in amino acid interactions and capsid stability performed using crystal structure of full assembled capsid on DynaMut server.

**Figure supplement 3.** VP1-L97R and VP1-E167G mutations decrease capsid stability.

**Figure supplement 4.** Raw micrographs and 2D classes of viruses incubated with soluble scavenger receptor class B member 2 (SCARB2).

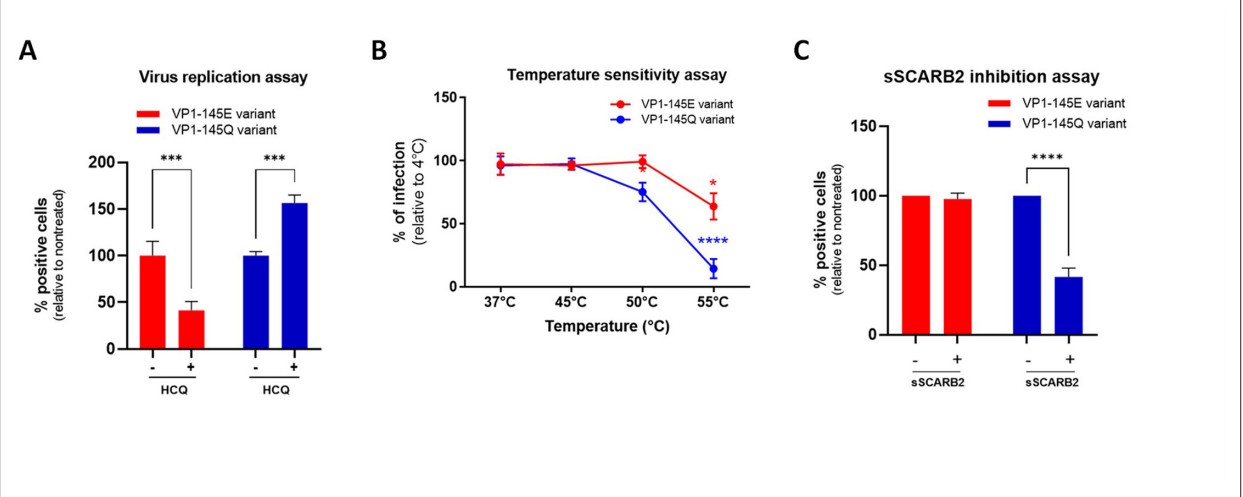

**Figure 6.** Heparan-sulfate-binding VP1-145Q variant exhibits resistance to hydroxychloroquine (HCQ) and higher sensitivity to soluble scavenger receptor class B member 2 (sSCARB2) inhibition and thermal stress. (**A**) Virus inhibitory assay with VP1-145 variants were performed with 25 µg HCQ on Vero cells (MOI 0.1). (**B**) For temperature sensitivity assays, VP1-145 variants were incubated at increasing temperature for 1 hr before inoculated onto Vero cells (MOI 0.5). (**C**) For sSCARB2 inhibition assay, VP1-145 variants were incubated 1 hr at 37 °C with 1 µg of soluble SCARB2 (sSCARB2) before infection of Vero cells (MOI 0.5). Infected cells were quantitated by immunostaining with anti-VP2 Ab at 24 hpi. Results are shown as % of virus-positive cells relative to nontreated control (A & C) or 4 °C treated control (**B**). Mean and S.E.M of biological triplicates are shown. Statistically significant differences (two-way ANOVA) are shown. **p<0.01, ***p<0.001, ****p<0.0001.

The online version of this article includes the following source data and figure supplement(s) for figure 6:

**Source data 1.** Related to *Figure 6A*.

**Source data 2.** Related to *Figure 6B*.

**Source data 3.** Related to *Figure 6C*.

**Figure supplement 1.** Human clinical strain variants exhibited similar hydroxychloroquine (HCQ) effects compared to MP4 variants.

**Figure supplement 1—source data 1.** Related to *Figure 6—figure supplement 1*.

One strength of our study lies in the fact that we were able to extend these results to another HS-dependent EV-A71 strain, the VP1-145Q variant. Given that these two HS-dependent variants share the characteristic of having incorporated a less acidic amino acid within the VP1 capsid protein, we hypothesized that an increase in positive charges within the capsid not only enhances affinity for HS but also alters capsid stability, consequently impacting the virus entry mechanism. In the same line, a thermostable EV-A71 variant (VP1-K162E, change of a basic to an acidic residue) isolated from serial passages at higher temperatures was shown to be less efficient at uncoating with poorer cell infectivity but more virulent in mice (*Catching et al., 2023*). Interestingly, this variant showed a more expanded conformation compared to the original non-mutated virus (*Catching et al., 2023*). While the thermostable variant showed no difference in binding to the SCARB2 receptor, the binding affinity to heparin was greatly reduced, an observation consistent with what we noticed for MP4. Additional experiments with cell-adapted, HS-binding viruses will help to define whether HS-binding is always associated with a loss of virion stability and whether these findings could even extend to other groups of viruses. Interestingly, mutations conferring similar *in vitro* phenotypes were observed for other enteroviruses such as rhinovirus A16 (RV-A16) and coxsackievirus B3 (CV-B3). For RV-A16, capsid mutations conferring resistance to endosomal acidification inhibitors also abrogated the need for acidic pH for uncoating. More importantly, these mutations were also associated with higher sensitivity to low pH, high temperatures, and binding to soluble receptors (*Murer et al., 2022*). For CV-B3, a fast-growing variant was shown to exhibit faster genome release and destabilized capsid, and this led to attenuated virulence in mice (*Lanahan et al., 2021*). These studies, which report a correlation between capsid instability, earlier uncoating, and attenuation in mouse models align with our findings. This suggests that amino acid changes affecting capsid stability can significantly impact various aspects of the virion and its life cycle, including sensitivity to environmental factors, receptor interactions, and infection rates.

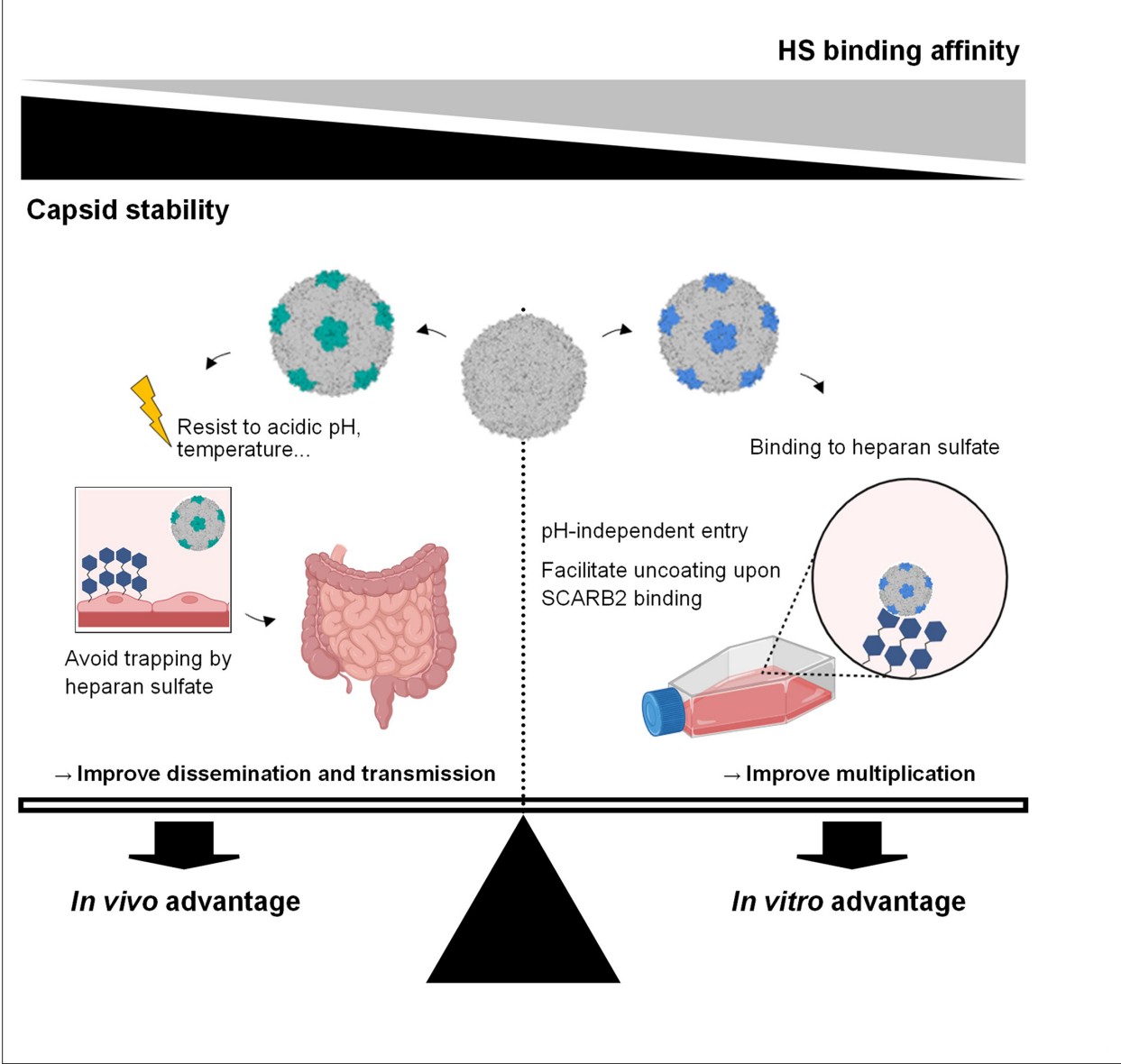

**Figure 7.** Seesaw model depicting the interplay between capsid mutations, heparan sulfate-binding, capsid stability as well as the resulting fitness changes in both *in vitro* and *in vivo* settings. Viruses undergo continuous mutations to optimize fitness across diverse environments. In cell culture, they adapt to attain an '*in vitro* advantage' by decreasing capsid stability while acquiring HS-binding capacity, consequently enhancing their infectivity. Conversely, during human infection, viruses adapt to secure an '*in vivo* advantage' by bolstering capsid stability, relinquishing heparan sulfate (HS)-binding capacity, and thereby evading viral trapping and resisting environmental stresses. Figure 7 was created with BioRender.com.

In the light of these published studies and our data, we propose the following model depicting the relationship between HS-binding, capsid stability, and viral fitness *in vitro* and *in vivo* (schematized in *Figure 7*). Viruses undergo mutations and positive selection to adapt to different environments (*LaTourrette and Garcia-Ruiz, 2022*; *Andino and Domingo, 2015*). EV-A71 can take advantage of the high plasticity of its capsid to optimize its fitness upon environmental changes. Many strains adapt to use HS *in vitro* due to the abundant expression of this attachment receptor in cell lines. To do so, they usually acquire additional positively charged amino acids within outward-facing VP1 domains proximal to the capsid fivefold axis. In addition to help the virus to attach to the cell surface and find SCARB2, we show here that these mutations concomitantly decrease virion stability. This further contributes to higher multiplication in cell lines by triggering uncoating rapidly after internalisation, within EE, without the need for acidic pH. Interestingly in our experiments, acidification inhibitors improved rather than inhibited viral fitness of MP4-97R/167G. Although additional experiments are

required to define the mechanism behind this observation, it could occur via the protection of virions that have not yet been uncoated when endosomes fuse with lysosomes. In this context, the absence of acidification would improve the chance of these virions to release their genome in the cytoplasm, while exposure to acidic pH would induce viral opening within the late endosomes. The situation may differ significantly *in vivo* as strong HS binders are attenuated. Koike and colleagues have demonstrated that strong binding to HS induces virus trapping *in vivo* (*Fujii et al., 2018*; *Kobayashi et al., 2018*; *Tee et al., 2019*; *Weng et al., 2023*). Our data suggest that the associated decreased stability may further contribute to viral attenuation. To be virulent *in vivo*, a non-enveloped virus must have a sufficiently stable capsid to resist unfavourable environmental conditions, both during dissemination within a host and during transmission between hosts. This last point is particularly important for EV-A71, which is transmitted via the fecal-oral route and must, therefore, resist the acidic pH of the stomach before reaching the intestinal mucosa, its main multiplication site. Capsid stability thus ensures that virus genome release occurs only in a proper environment, but in turn, renders the virus dependent on both SCARB2 and acidic pH for uncoating (*Yamayoshi et al., 2013*; *Dang et al., 2014*).

Of note, although strong HS-binder is clearly attenuated in mice, the situation in humans is more puzzling as strains with Q or G at position VP1-145 have been associated with severe neurological cases, and the methodology used in those studies excluded the emergence of mutation during cell culture (*Chang et al., 2012*; *Liu et al., 2014*). Experimental work in human-relevant tissue culture models also suggests that binding to HS provides an advantage to the virus (*Aknouch et al., 2023*; *van der Sanden et al., 2018*; *Tseligka et al., 2018*). Furthermore, we previously showed that intermediate HS-binding affinity can result in increased virulence even in mouse models (*Weng et al., 2023*). We are currently investigating if there is a lack of trapping and/or limited impact on viral stability under these conditions. To conclude, to reach optimal fitness *in vitro* and *in vivo*, the virus needs to find the correct balance between HS binding and capsid stability (*Figure 7*). Our study improves the current knowledge of the mechanism behind the *in vivo* attenuation of cell culture-adapted viruses. It also opens the doors to new antiviral strategies targeting endosomal acidification as well new principles for vaccine design based on attenuated-acid-independent variants to help combat EV-A71.

# Materials and methods

**Key resources table**

| Reagent type (species) or resource | Designation | Source or reference | Identifiers | Additional information |
|---|---|---|---|---|
| Antibody | Anti-enterovirus 71 Ab (mouse monoclonal) | Merck Millipore | MAB979; RRID:AB_95300 | IF (1:1000) RNAscope (1:100) |
| Antibody | Anti-dsRNA monoclonal antibody J2 (mouse monoclonal) | Scicons | RRID:AB_2651015 | IF (1:500) |
| antibody | Anti-LAMP1 antibody (rabbit monoclonal) | Cell Signaling | D2D11, RRID:AB_2687579 | IF (1:100) |
| Antibody | Anti-EEA1 antibody (goat polyclonal) | Santa Cruz | sc-6414, RRID:AB_640035 | IF (1:100) |
| Antibody | Goat anti-Mouse IgG (H+L) Highly Cross-Adsorbed Secondary Antibody, Alexa Fluor 488 (goat polyclonal) | Thermo Fisher Scientific | A-11029; RRID:A-11029 | IF (1:2000) RNAscope (1:200) |
| Antibody | Goat anti-Mouse IgG (H+L) Highly Cross-Adsorbed Secondary Antibody, Alexa Fluor 594 (goat polyclonal) | Thermo Fisher Scientific | A-11032; RRID:AB_2534091 | IF (1:2000) |
| Strain, strain background (virus) | MP4 | *Cagno et al., 2019* | GenBank: JN544419 | Mouse-adapted virus |
| Strain, strain background (virus) | HU-97L | *Tee et al., 2021* | GenBank: EU414331 | Clinical strain |

*Continued on next page*

*Continued*

| Reagent type (species) or resource | Designation | Source or reference | Identifiers | Additional information |
|---|---|---|---|---|
| Strain, strain background (virus) | IEQ | *Kobayashi and Koike, 2020* | GenBank: AF316321 | |
| Strain, strain background (virus) | IEE | *Kobayashi and Koike, 2020* | GenBank: AF316321 | |
| Chemical compound, drug | Hydroxychloroquine | Tocris | 747-36-4 | |
| Chemical compound, drug | Bafilomycin A1 | InvivoGen | 88899-55-2 | |
| Chemical compound, drug | Sodium chlorate | Sigma | 7775-09-9 | |
| Chemical compound, drug | Neutral red | Sigma Aldrich | 553-24-2 | |
| Chemical compound, drug | Heparinase III | Amsbio | 37290-86-1 | |
| Chemical compound, drug | Puromycin | InvivoGen | 58-58-2 | |
| Chemical compound, drug | LysoTracker Deep Red | Thermo Fisher Scientific | LysoTracker Deep Red | |
| Chemical compound, drug | SYBR green II RNA gel stain | Thermo Fisher Scientific | SYBR green II RNA gel stain | |
| Recombinant protein | Recombinant Human LIMPII/SR-B2 Fc Chimera Protein, CF | Bio-Techne | 1966-LM | |
| Commercial kit or assay | RNAscope V-EV71-C1 probe | Biotechne | 1087481-C1 | |
| Commercial kit or assay | RNase P housekeeping gene | Thermo Fisher Scientific | 4316861 | |
| Commercial kit or assay | Lipofectamine 2000 | Thermo Fisher Scientific | 11668019 | |
| Commercial kit or assay | Superscript II reverse transcriptase | Thermo Fisher Scientific | 18064022 | |
| Commercial kit or assay | Random hexamer primers | Roche | 11034731001 | |
| Commercial kit or assay | Platinum Taq DNA Polymerase, DNA-free | Thermo Fisher Scientific | 15966025 | |
| Commercial kit or assay | TSA Vivid 570 kit | Tocris | 7526 | |
| Commercial kit or assay | E.Z.N.A. Viral RNA kit | Omega Bio-Tek | R6874-02 | |
| Commercial kit or assay | RNAscope Multiplex Fluorescent V2 | Biotechne | 323270 | |
| Commercial kit or assay | RNA-Protein Co-detection Ancillary Kit | Biotechne | 323180 | |
| Commercial kit or assay | KAPA SYBR FAST One-Step qRT-PCR Kits | Kapa Biosystems | KK4650 | |
| Commercial kit or assay | Nano-Glo Luciferase Assay System | Promega | N1110 | |
| Commercial kit or assay | Thiazolyl Blue Tetrazolium Bromide (MTT) | Merck | M5655 | |

*Continued on next page*

*Continued*

| Reagent type (species) or resource | Designation | Source or reference | Identifiers | Additional information |
|---|---|---|---|---|
| Commercial kit or assay | CyQUANT LDH Cytotoxicity Assay | Thermo Fisher Scientific | C20300 | |
| Cell line (*Cercopithecus aethiops*) | Vero cells | ATCC, USA | RRID:CVCL_0059 | |
| Cell line (*Homo sapiens*) | Rhabdomyosarcoma (RD) cells | ATCC, USA | RRID:CVCL_1649 | |
| Cell line (*Homo sapiens*) | RD-SCARB2-KO | Caroline Tapparel **Yamayoshi et al., 2013** | | |
| Cell line (*Homo sapiens*) | RD-ΔEXT1+hSCARB2 | Satoshi Koike **Kuronita et al., 2002** | | |
| Sequence-based reagent | RT-qPCR assay primer/for (Entero/Ge/08 assay) | **Nishimura et al., 2024** | PCR primers | 5'-GCTGCGYTGGCGGCC-3' |
| Sequence-based reagent | RT-qPCR assay primer/Rev (Entero/Ge/08 assay) | **Nishimura et al., 2024** | PCR primers | 5'-GAAACACGGACACCCAAAGTAGT-3' |
| Sequence-based reagent | RT-qPCR assay primer/probe (Entero/Ge/08 assay) | **Nishimura et al., 2024** | PCR primers | 5'-CTCCGGCCCCTGAATGYGGCTAA-3' |
| Recombinant DNA reagent | EV-A71/MP4 (Genbank accession number: JN544419; subgenogroup C2) | Jen-Reng Wang **Guo et al., 2022** | | |
| Recombinant DNA reagent | IEQ (Genbank accession number: JN544419: AF316321; subgenogroup B4) | Jen-Reng Wang **Kobayashi and Koike, 2020** | | |
| Recombinant DNA reagent | IEE (Genbank accession number: JN544419: AF316321; subgenogroup B4) | Jen-Reng Wang **Kobayashi and Koike, 2020** | | |
| Recombinant DNA reagent | eGFP-Rab5a WT | Pierre-Yves Lozach **Dang et al., 2014** | | |
| Recombinant DNA reagent | eGFP-Rab5a S34N | Pierre-Yves Lozach **Dang et al., 2014** | | |
| Recombinant DNA reagent | eGFP-Rab5a Q79L | Pierre-Yves Lozach **Dang et al., 2014** | | |
| Software | Geneious 10.2.16 | https://www.geneious.com | https://www.geneious.com | |
| Software | ImageXpress Micro XL | Molecular Devices | Molecular Devices | |
| Software | GraphPad Prism 9 | https://www.graphpad.com/scientific-software/prism/ | https://www.graphpad.com/scientific-software/prism/ | |
| Software | UCSF Chimera (version 1.13.1) | https://www.cgl.ucsf.edu/chimera/ | https://www.cgl.ucsf.edu/chimera/ | |

## Chemical reagents

Chemical reagents used in this study were listed as follows: hydroxychloroquine (Tocris), bafilomycin-A1 (InvivoGen), sodium chlorate (NaClO$_3$, Sigma-Aldrich), neutral red (Sigma Aldrich), and LysoTracker Deep Red (Thermo Fisher Scientific).

## Cell lines and virus

Vero (monkey kidney; ATCC CCL-81) and human rhabdomyosarcoma cells (RD; ATCC no.: CCL-136) were propagated in Dulbecco's Modified Eagle Medium (DMEM) and GlutaMAX (31966021, Thermo

Fisher Scientific) containing 10% fetal bovine serum (FBS). RD-SCARB2-KO (*Tseligka et al., 2018*) and RD-ΔEXT1+hSCARB2 (*Kobayashi et al., 2020*) cells were maintained in DMEM supplemented with 10 µg/ml puromycin (58-58-2, InvivoGen). All infected cells were maintained in media supplemented with 2.5% FBS. All cells were maintained at 37 °C in 5% $CO_2$. Viruses used in this study including MP4, MP4-97R/167 G (Genbank accession number: JN544419; subgenogroup C2), IEQ (EV-A71 VP1-145Q variant; Genbank accession number: AF316321; subgenogroup B4), IEE (EV-A71 VP1-145E variant), HU-97L (Genbank accession number: EU414331) and HU-97R/167 G strains were prepared as previously described (*Tee et al., 2019*; *Cordey et al., 2012*; *Tseligka et al., 2018*; *Weng et al., 2023*). For the Nluc reporter virus, the Nluc gene was inserted between 5′ UTR and VP4 of the virus as previously described (*Tan et al., 2016*). Viruses were generated in RD-ΔEXT1+hSCARB2 cells (*Kobayashi et al., 2020*), propagated for an additional passage, and used as working stocks. All virus stocks were sequenced for confirmation (Fasteris) prior to experiments.

## Plasmids

Plasmids encoding eGFP-Rab5a and eGFP-Rab5a Q79L are kind gifts from Pierre-Yves Lozach (University Claude Bernard Lyon 1). Both IEQ and IEE plasmids (Genbank accession number: JN544419: AF316321; subgenogroup B4) strains are kind gifts from Yoke Fun Chan (University of Malaya).

## Virus inhibitory assay and time-of-addition assay

For virus inhibitory assay, cells were pre-treated either with drugs for 1 hr at 37 °C. Viruses (MOI 0.1) were inoculated onto cells in the presence of drugs for 1 hr at 37 °C. Upon infection, inocula were removed and cells were rinsed thoroughly with phosphate-buffered saline buffer (PBS) before being incubated with fresh media up to 24 hpi. For the time-of-addition assay, HCQ was either pretreated (–1 hpi), introduced during virus infection (0hpi) or introduced onto cells at post-infection (1, 2, and 3 hpi) for 1 hr. After incubation, cells were rinsed with PBS before being loaded with maintenance media and incubate for up to 24 hpi. For both assays, infected cells were fixed for immunofluorescence staining for virus-positive cell detection.

## Virus binding and replication assay

All the experiments were done on Vero cells seeded in 96-wells plates. For virus binding assay, cells were incubated with $1\times10^8$ RNA copy number/ml virus for 1 hr at 4 °C. The inocula were removed and rinsed with cold PBS twice, and then subjected to cell lysis for RNA extraction and qRT-PCR quantitation. For the virus replication assay, cell monolayers were incubated with the virus for 1 hr at 37 °C. The inocula were removed, rinsed with PBS, and then further incubated up to 24 hpi at 37 °C. Infected cells were lysed and viral RNA was quantified by qRT-PCR.

## RNA extraction and real-time quantitative polymerase chain reaction

Viral RNA was extracted using the OMEGA E.Z.N.A. Viral RNA kit (R6874-02) according to the manufacturer's instructions. In brief, 100 µl of lysis buffer was added directly onto infected cells with supernatant, and 50 µl out of 200 µl lysate was collected to proceed for RNA extraction. RT-qPCR was performed based on quantitative Entero/Ge/08 assay (forward primer GCTGCGYTGGCGGCC, reverse primer GAAACACGGACACCCAAAGTAGT, and probe CTCCGGCCCCTGAATGYGGCTAA) as previously described (*Filipe et al., 2022*) using the QuantiTect Probe RT-PCR Kit (Qiagen, Switzerland) in 7500 Real-Time PCR System (Applied Biosystems).

## Heparan sulfate removal

Both enzymatic and chemical methods, heparinase assay and sodium chlorate ($NaClO_3$) treatment respectively, were used to cleave HS from the cell surface. For the heparinase assay, the cells were first rinsed with PBS and then incubated with 3.5 mIU/ml of heparinase III (AmsBio) diluted in 0.1 M sodium acetate pH 7.0, 1 mM calcium acetate, and 0.2% BSA for 1 hr at 37 °C. Meanwhile, mock-treated cells were incubated with heparinase buffer. Upon incubation, cells were washed twice prior to virus infection. For $NaClO_3$ treatment, cells were propagated in the presence of 30 mM $NaClO_3$ at least one passage before the experiment. The cells were also pre-seeded in media supplemented with $NaClO_3$ and incubated overnight before the experiment.

## Immunofluorescence and confocal imaging

To detect infected cells, cells were fixed with absolute methanol (Sigma Aldrich) at room temperature for 10 min and then incubated with a blocking buffer consisting of 5% BSA (PanReac Applichem) and 0.05% TritonX-100 (PanReac Applichem) for 20 min. Fixed cells were first incubated with anti-EV-A71 capsid monoclonal antibody MAB979 (1:1000; Sigma) for 1 hr at 37 °C and then with Alexa Fluor 488-conjugated secondary antibodies (1:2000; Thermo Fisher Scientific) dissolved in DAPI solution for 1 hr at 37 °C. To detect dsRNA, infected cells were fixed with 4% paraformaldehyde (Santa Cruz) and incubated with a blocking buffer, cells were incubated with anti-dsRNA monoclonal antibody J2 (1:500; Scicons) for 1 hr at 37 °C and then with Alexa Fluor 594-conjugated secondary antibodies (1:2000; Thermo Fisher Scientific) dissolved in DAPI solution for 1 hr at 37 °C. Stained cells were acquired using ImageXpress Pico (Molecular Devices) and percentages of positive cells were determined using Cell-ReporterXpress software. Images acquired were also processed using Imaris software and displayed as double-positive cells. For confocal imaging, immunofluorescence staining was performed with EE and lysosomes were stained using anti-EEA1 (1:100; Santa Cruz) and anti-LAMP1 (1:100; Cell Signalling), respectively, for 1 hr at 37 °C and then with Alexa Fluor 488-conjugated secondary antibodies (1:200) dissolved in DAPI solution for 1 hr at 37 °C. The stained slides were mounted under a coverslip (Hecht Assistent) with Fluoromount G mounting medium (Southern Biotech) and analyzed using Zeiss LSM 800 confocal microscopy.

## RNAscope FISH detection and colocalization experiments

For FISH, cells were seeded on Nunc LabTek II chamber slides (Thermo Fisher Scientific) and fixed with 4% paraformaldehyde. To detect viral RNA in infected cells, fixed cells were processed for RNAscope FISH using RNAscope Multiplex Fluorescent V2 assay (Biotechne) according to the manufacturer's protocol. In brief, the cells were hybridized with a V-EV71-C1 probe (Biotechne) at 40 °C for 2 hr and then the signals were revealed using TSA Vivid 570 kit (Tocris). The slides were then incubated with blocking buffer and incubated with MAB979 (1:100) at 4 °C overnight, followed by incubation with Alexa Fluor 488-conjugated secondary antibodies (1:200) at room temperature for 30 min. After incubating with DAPI for 30 s, the stained slides were mounted under a coverslip with mounting medium and analyzed using Zeiss LSM 800 confocal microscopy. Images were acquired and analyzed using ZEN 3.2 software.

## Luciferase assay

Enterovirus-A71 nanoluciferase (Nluc) reporter particles were used to study virus replication bypassing cell entry in presence and absence of the drug. Briefly, the reporter virus plasmid was linearized and *in vitro* transcribed to generate RNA using the T7 RiboMax Express Large Scale RNA Production System (Promega). Transcribed RNA was purified using RNeasy Mini Kit (Qiagen) and then transfected in RD cells using Lipofectamine 2000 (Thermo Fisher Scientific). At certain timepoints, cell supernatants were harvested for luciferase activity detection using the Nano-Glo Luciferase Assay System kit (Promega) on the Glomax Multi-Detection System (Promega).

## Neutral red uncoating assay

To generate neutral red (NR)-labeled viruses, virus stocks were propagated in cells in presence of 5 µg/ml neutral red (Aldrich). The virus stocks were harvested at 3 dpi and titered. For uncoating assay, NR-labeled viruses were infected at 37 °C for 1 hr in the dark then washed twice with PBS and loaded with FluoroBrite DMEM (Thermo Fisher Scientific) supplemented with 2.5% FBS. At certain timepoints, infected cells were exposed to light for 30 min and then allowed to incubate for up to 24 hpi. Infected cells were analyzed using immunofluorescence as stated earlier.

## Virus infection in Rab5a-transfected cells

Vero cells ($1.5 \times 10^6$) were transfected with 25 µg of Rab5a-eGFP plasmids using Lipofectamine 3000 (Thermo Fisher Scientific). The next day, transfected cells were harvested, resuspended in buffer (PBS, 2 nM EDTA, 1% BSA), and subjected to fluorescence-activated flow cytometry (FACS) on S3 Cell Sorter (Biorad). EGFP-positive cells were sorted, collected, and then further propagated at least 1 d before virus infection. For virus infection, cells were infected with virus (MOI 1.5) for 1 hr at 37 °C. The

inocula were removed, rinsed with PBS, and cells were further incubated up to 7 hpi at 37 °C. Cells were then stained with anti-dsRNA as described above.

## Temperature sensitivity assay and shSCARB2 inhibition assay

Viruses (MOI 0.5) were incubated at different temperatures (4 °C, 37 °C, 45 °C, 50°C, and 55°C) for 1 hr. Upon incubation, viruses were immediately transferred onto ice for cooling down before inoculated onto cells for 1 hr at 37 °C. Cells were washed and allowed to incubate in maintenance media up to 24 hpi before virus-positive cell detection using immunofluorescence. For SCARB2 inhibition assay, viruses were incubated with 1 μg of soluble recombinant human SCARB2-FC chimera protein (bio-techne) at 37 °C for 1 hr. The mixture was then inoculated onto cells at 37 °C for 1 hr. Upon incubation, cells were washed, and allowed to incubate in maintenance media up to 7 hpi before lysing the cells for viral RNA quantitation.

## Electron microscopy (EM)

For structural analyses, virus stocks were first inactivated by formaldehyde treatment. Formaldehyde at 100 μg/ml final concentration was added to the virus stock and incubated at 37 °C for 3 d. Inactivated viruses were purified through a 30% sucrose cushion at 32,000 rpm in SW32 Ti rotor (Beckman Coulter) for 14 hr at 4 °C, followed by sedimentation through a discontinuous 20–45% (w/v) sucrose at SW41 Ti rotor (Beckman Coulter) for 12 hr at 4 °C. The purified stocks were then subjected to HiPrep 16/60 Sephacryl S-500 HR column (Sigma Aldrich) with 25 mM Tris-HCl +150 mM NaCl (pH 7.5) as the running buffer. Fractions corresponding to EV A71 particles were pooled and concentrated to 0.3–1.1 mg/mL using Amicon Ultra centrifugal filter units with 100 kDa cutoff (Millipore Sigma). For pH-based assays we prepared Tris-Acetate-based buffers at pH 5 and pH 7.5. The buffers comprised 150 mM NaCl and a 100 mM mix of Tris base and acetic acid at the ratio necessary to reach the desired pH. Each EV A71 variant was diluted to 100 μg/ml in the two buffer and incubated for 30 min. Following incubation, the samples were applied onto negative stain EM grids (Cat # CF300-Cu-50, Electron Microscopy Sciences). Prior to sample application the grids were glow discharged for 30 s. 2% solution of uranyl formate was used for staining. The grids were imaged on a Talos L120C G2 microscope (Thermo Fisher Scientific) running at 120 kV and featuring the CETA 4 k camera. EPU software from Thermo Fisher Scientific was used for data acquisition, and all data processing was performed in the cryoSPARC package (*Punjani et al., 2017*). Each dataset comprised 100–200 micrographs, and 2000–10,000 virus-corresponding particles. Particles were extracted from micrographs and subjected to 2D classification. 3D reconstruction was performed using Ab initio algorithm with icosahedral symmetry imposed. For experiments at pH 5 where the expanded viral particles were observed, the Ab initio reconstruction was performed without the imposition of symmetry (i.e. C1). For imaging the virus in complex with SCARB2, the virus was combined with 1:1 molar ratio of SCARB2 to the viral P1 protomer and diluted to 100 μg/ml with Tris-Acetate-based buffer at pH 5 or pH 7.5 as mentioned earlier. Upon incubation, 5% glycerol was added to each sample to minimize aggregation. The samples were then applied to glow discharged grids and stained with 2% uranyl acetate, The imaging and analysis were carried out as mentioned above. Empty capsid quantification was performed on the level of 2D class-averages using the degree of staining in the center of viral particle as the main discriminatory factor.

## Computational analysis of virus capsid protein structure stability

To assess the virus capsid protein structure stability, EV-A71 crystal structures with PDB ID of 3J22 and 4AED were used for MP4 and VP1-145 variants, respectively. I-mutant 2.0 server (*Capriotti et al., 2005*) was used to predict the free energy stability change upon introduction of mutation into virus capsid VP1 protein. Visualization of mutational effects on interatomic interactions and prediction of molecule flexibility were performed on DynaMut server (*Rodrigues et al., 2018*).

The electrostatics changes towards mutations, and the visualization of their effect on the capsid surface were studied using the Adaptive Poisson–Boltzmann Solver (APBS) (*Bank and Holst, 2003*; *Jurrus et al., 2018*). We used a linear Poisson-Boltzmann equation with a 'Single Debye-Hueckel' boundary condition. Protein and solvent dielectric constants were set to 2.0 and 78.0, respectively, with a cubic B-spline discretization for charge distribution and a smoothed molecular surface for dielectric and ion-accessibility coefficients. We included monovalent ions at a concentration of 0.15 M, with radii

of 2.0 Å for positive ions and 1.8 Å for negative ions. The analysis was performed and the potential maps were generated using the APBS PyMol (*Schrodinger LLC, 2010*) plugin.

## Schematic diagram and statistical analysis

All experiments were performed in biological triplicate, unless otherwise stated. All schematic diagrams and illustrations were created via BioRender.com. All data and statistical analyses were generated using GraphPad Prism 9. All drug treatment experiments were analyzed with one-way and two-way ANOVA. For dose-dependent inhibitory assay, area under curve (AUC) was calculated and analyzed using one-way ANOVA. Degree of colocalization of virus capsid and vRNA in individual cells was measured using Mander's overlap coefficient calculation in ZEN 3.2 software. Data were presented as mean ± SEM. $*p<0.05$, $**p<0.01$, $***p<0.001$, $****p<0.0001$. and not significant (n.s.).

## Acknowledgements

This work was funded in part by the Swiss national foundation (Grant No 501100001711-10000300 and 501100001711-184777 to CT) and by the University of Geneva (Salary to HKT). We would like to thank Prof Satoshi Koike and Dr Kyousuke Kobayashi from Tokyo Metropolitan Institute of Medical Science, Japan for providing RD-ΔEXT1+hSCARB2 cells, Prof Jen-Ren Wang from National Cheng Kung University, Taiwan for providing infectious clone plasmids EV-A71/MP4, Prof Pierre-Yves Lozach from Université Claude Bernard Lyon 1 for providing plasmids encoding eGFP-Rab5a and eGFP-Rab5a Q79L, Prof Yoke Fun Chan from University of Malaya for providing IEQ and IEE infectious clone plasmids. We would also like to acknowledge Jessica Swanson, Dr Natalie Kingston and Prof Nicola Stonehouse for giving advice and guidance about virus purification. Electron microscopy data was collected at the Interdisciplinary Centre for Electron Microscopy (CIME) at EPFL with assistance from Davide Demurtas, PhD. Electron microscopy data was processed using the computational infrastructure provided by the IT department of the School of Life Sciences (SV-IT) at EPFL. The authors express sincere gratitude to the CIME and SV-IT personnel for their contribution.

## Additional information

### Funding

| Funder | Grant reference number | Author |
| --- | --- | --- |
| Swiss national foundation | Grant No 501100001711-10000300 and 501100001711-184777 | Caroline Tapparel |

The funders had no role in study design, data collection and interpretation, or the decision to submit the work for publication.

### Author contributions

Han Kang Tee, Conceptualization, Data curation, Formal analysis, Validation, Investigation, Visualization, Methodology, Writing - original draft, Writing – review and editing; Simon Crouzet, Data curation, Software, Formal analysis, Methodology, Writing – review and editing; Arunima Muliyil, Data curation, Formal analysis, Investigation; Gregory Mathez, Data curation, Formal analysis, Validation, Investigation, Methodology, Writing – review and editing; Valeria Cagno, Conceptualization, Data curation, Formal analysis, Validation, Investigation, Methodology; Matteo Dal Peraro, Software, Formal analysis, Validation, Methodology, Writing – review and editing; Aleksandar Antanasijevic, Data curation, Formal analysis, Investigation, Methodology, Writing – review and editing; Sophie Clément, Conceptualization, Data curation, Formal analysis, Validation, Methodology, Writing – review and editing; Caroline Tapparel, Conceptualization, Resources, Supervision, Funding acquisition, Validation, Methodology, Project administration, Writing – review and editing

### Author ORCIDs

Han Kang Tee ⓘ https://orcid.org/0000-0002-8975-0945
Gregory Mathez ⓘ https://orcid.org/0000-0002-4453-7649

Valeria Cagno [iD] https://orcid.org/0000-0002-5597-334X
Aleksandar Antanasijevic [iD] https://orcid.org/0000-0001-9452-8954
Sophie Clément [iD] http://orcid.org/0000-0003-1348-4887
Caroline Tapparel [iD] https://orcid.org/0000-0002-0411-6567

### Decision letter and Author response
Decision letter https://doi.org/10.7554/eLife.98441.sa1
Author response https://doi.org/10.7554/eLife.98441.sa2

## Additional files

### Supplementary files
Supplementary file 1. Predicted Gibbs free energy change value (ΔΔG) was computed using I-mutant 2 server with calculation formula and an indication of protein structure stabilization as shown: Predicted Gibbs free energy change value (ΔΔG): ΔG (mutated protein) - ΔG (WT) in kcal/mol. ΔΔG<0: Destabilizing mutation. ΔΔG>0: Stabilizing mutation.

MDAR checklist

### Data availability
The authors confirmed that all the raw data used to plot the graphs and figures were included in the source data files.

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
