## [Editor Report]

This is important work, correlating capsid stability with mutations that promote heparan sulfate binding. The data are solid and support the claims.

---

## [Decision Letter]

**Decision letter after peer review:**

Thank you for submitting your article "Virus adaptation to heparan sulfate comes with capsid stability tradeoff" for consideration by *eLife*. Your article has been reviewed by 3 peer reviewers, one of whom is a member of our Board of Reviewing Editors, and the evaluation has been overseen by John Schoggins as the Senior Editor.

Essential revisions (for the authors):

In particular, these are the five more pressing points that should be addressed

1) One of the weaknesses is a lack of explanation for the mechanism(s) of capsid destabilization conferred by overall increased positive charge. The manuscript should be enhanced by additional biophysical and computational studies to establish the link between the link between increase in net positive charge and stability.

2) The inclusion of RT-qPCR and/or plaque assays as supplementary data will help strengthen the findings.

3) There is a need for a better analysis of the influence of overall charge alterations, or individual mutations, on subunit interaction or particle conformation.

4) It is not clear how different this strain is from circulating EV-A71 strains and the relevance of these findings to the human situation is questionable.

5) The DynaMut Server data must be corroborated by analyzing how the mutations in their attenuated strain affect the protein interactions in the virion by looking at the available Cryo-EM data for Enterovirus A71.

Also, please address all the points in the section "Recommendations for the Author".

*Reviewer #1 (Recommendations for the authors):*

Strengths

1. The entry assays determining if the mutant requires acidification are highly robust, well-thought-out, and convincing. These assays are very interesting because they use drugs that affect the endosomal pH and different cell lines that express different amounts of HS.

2. The neutral red assays coupled with FISH to test uncoiling are also very robust and convincing. These data are consistent with the entry data.

3. The results of chemical blockage of endosomal acidification were further corroborated by using the Rab5a-Q79L; most people would have stopped with the HCQ data, but the fact that they used this mutant supports the hypothesis.

4. The thermal treatment of the viruses is the best way to show if their hypothesis is correct and if the data from those experiments is consistent with the whole idea.

Weaknesses

1. The DynaMut Server data could have been corroborated by analyzing how the mutations in their attenuated strain affect the protein interactions in the virion by looking at the available Cryo-EM data for Enterovirus A71.

2. Negative TEM staining is not the best way to obtain 2D and 3D reconstructions (nonetheless, they are perfect).

3. The methodology does not explain the number of biological replicas performed in most experiments (although it is described in the figure's legends).

4. Vero cells are challenging to transfect, so it is ideal to know the transfection efficiency of the Rab5a-Q79L experiments.

*Reviewer #3 (Recommendations for the authors):*

The manuscript should be enhanced by additional biophysical and computational studies to establish the link between the link between increase in net positive charge and stability. Also, any distinction between the entry pathway of variants needs to be considered.

The entry pathway of the L97R/E167G variant, in light of enhanced infectivity under elevated endosomal pH, requires more probing. How would close to neutral pH protect uncoated particles?

Are the authors convinced that MP4 and the L97R/E167G variant show the exact same pattern for cellular surface association? Has SCRAB2 association *in vitro* or cellular association been measured/quantified by any biophysical method? Also, is it possible that the variants enter cells through some other pathway – macropinocytosis, for example?

The MP4 particle and the variants can be subjected to Differential Scanning Calorimetry (DSC) to understand the difference in stability and for semi-quantification of the difference among variants. Does progressive increase in overall positive charge further decrease the stability of variants? Likewise, the in silico stability analysis should be expanded to analyze subunit interaction.

The effect of both receptor binding and low pH on the morphology of the particles *in vitro* should be tested. The MP4 particles do not seem to form A or B particles even at pH 5.0, will this happen with a conjunction of receptor binding and low pH?

The capsid alterations in the reconstructions require adequate explanation. The images of the variant at low pH indicate asymmetric conformational change – under such circumstances, application of icosahedral symmetry may cause misleading density interpretation. The 5-fold axes appears to be altered in the variant at low pH – can this be explained in the context of uncoating-related alterations?

Do the mutations correlate with calcium ion coordination, subunit interfaces or pocket factor binding in the context of the capsid structure? Do these residues influence the conformation of the dynamic regions of the capsid – the VP4 peptide or the N-terminus of VP1 – needed for endosomal membrane rupture?

*Reviewer #4 (Recommendations for the authors):*

A few specific comments/changes that will improve the manuscript are provided below:

1. Taxonomically, it should be written as Enterovirus A71. The hyphen (-) in line 46 between Enterovirus and A71 must be removed.

2. A few references are missing in the introduction or some statements may be based on earlier or later references but this is not immediately evident (e.g. line 45 and line 47).

3. Please adjust line 52 and others where SCARB2 is suggested to be important for internalization. It can be referred to as an entry receptor as uncoating is demonstrably an entry step. Alternatively, refer to it as the uncoating receptor.

4. Lines 59-63 are a summary of another study but given the lack of contextualization to the human data this should be removed and/or the human data should be better incorporated in the introduction.

5. Please add the quantification of Figure 1B.

6. Line 153 states that MP4 uncoating was completely blocked but this is not true from Figure 3B. Please adjust.

7. The MOI used for the different experiments is not clear. Please mention this in the figure captions.

8. Please perform statistical analysis on Figure 2F.

9. Please provide the individual panels for some of the fluorescence images such as 1B and 3C.

10. Please provide a set of representative images corresponding to Figure 4D.

11. In the discussion, it is stated that existing literature aligns with the findings. However, this is existing literature on mice but not humans which is in contrast. Please adjust this section based on the comments above.

12. In line 330, it is stated that the pH of the gastrointestinal tract is acidic. However, this is not true for the entirety of the GI tract and some parts are close to neutral pH.

---

## [Author Response]

Essential revisions (for the authors):Reviewer #1 (Recommendations for the authors):Strengths1. The entry assays determining if the mutant requires acidification are highly robust, well-thought-out, and convincing. These assays are very interesting because they use drugs that affect the endosomal pH and different cell lines that express different amounts of HS.2. The neutral red assays coupled with FISH to test uncoiling are also very robust and convincing. These data are consistent with the entry data.3. The results of chemical blockage of endosomal acidification were further corroborated by using the Rab5a-Q79L; most people would have stopped with the HCQ data, but the fact that they used this mutant supports the hypothesis.4. The thermal treatment of the viruses is the best way to show if their hypothesis is correct and if the data from those experiments is consistent with the whole idea.

We thank the reviewer for his thorough review and for raising important points that enabled us to improve the manuscript. We also appreciate the recognition of the importance of our study.

Weaknesses1. The DynaMut Server data could have been corroborated by analyzing how the mutations in their attenuated strain affect the protein interactions in the virion by looking at the available Cryo-EM data for Enterovirus A71.

We have performed additional computational analysis to study the impact of the mutations for protein-protein interactions. The outcomes of our analysis using the Adaptive Poisson–Boltzmann Solver (APBS), have been incorporated in the Figure 5—figure supplement 1 and discussed in the text (Page 6, line 208). This new result is in line with our previous assumptions and shows that a change in electrostatic potential is induced around the interaction region.

Result section page 6, line 208: “We analyzed the impact of VP1-97R and VP1-167G mutations on their respective local environments at pH 5 and pH 7 with the Adaptive Poisson-Boltzmann Solver (APBS)^33,34^. As displayed in Figure 5—figure supplement 1, both L97R and E167G mutations are inducing electrostatic changes at the surface of the capsid within the region of interest and the changes are particularly significant at pH 5. Furthermore, analysis with DynaMut server^35^ reveals that the two mutations affect interaction networks (Figure 5—figure supplement 2). The VP1-L97R mutation is predicted to reduce hydrophobic interactions between amino acid 97 and its neighbors VP1-245Y and VP1-246P while the VP1-E167G mutation causes a loss in hydrogen bonding capacity to VP1-165S and reduces the net negative charge. This is consistent with analyses of vibrational entropy change (Figure 5—figure supplement 3), indicating that the presence of two mutations results in enhanced local dynamics, which has previously been correlated with reduced capsid stability^36,37^. Consistently, the predictions of Gibbs free energy change (ΔΔG) induced by these mutations further support that both mutations induce destabilization of the capsid structure, regardless of pH and temperature (Figure 5-table supplement 1).”

2. Negative TEM staining is not the best way to obtain 2D and 3D reconstructions (nonetheless, they are perfect).

Indeed, negative-stain EM is a method whose resolution does not allow the interpretation of the molecular details of viral capsid assembly (i.e., visualization of individual VPs or relaxation of atomic models). However, in this manuscript, the reconstructed 3D maps are only used to evaluate the size and shape of viral particles. Our interpretation of the data does not go beyond the resolution limit of this method (~20 Å). Further, map reconstruction was performed using the Ab-initio algorithm and without providing any reference model (with or without imposition of icosahedral symmetry, depending on the sample) (see new version of Figure 5B). Therefore, we believe that the use of reconstructed 3D maps from negative stain EM images is appropriate under these conditions.

3. The methodology does not explain the number of biological replicas performed in most experiments (although it is described in the figure's legends).

We have now clarified this point in the method section (Page 19, line 552): “All experiments were performed in biological triplicate, unless otherwise stated.”

4. Vero cells are challenging to transfect, so it is ideal to know the transfection efficiency of the Rab5a-Q79L experiments.

transfected cells, we sorted the transfected cells (GFP-positive) using fluorescence-activated cell sorting (FACS) and ensured that the same number of transfected cells was plated for each experiment. This is detailed in the Methods section on page 17, line 486:

“Virus infection in Rab5a-transfected cells

Vero cells (1.5 ×106) were transfected with 25µg of Rab5a-eGFP plasmids using Lipofectamine 3000 (Thermo Fisher Scientific). The next day, transfected cells were harvested, resuspended in buffer (PBS, 2nM EDTA, 1% BSA), and subjected to fluorescence-activated flow cytometry (FACS) on S3 Cell Sorter (Biorad). EGFP-positive cells were sorted, collected and then further propagated at least one day before virus infection. For virus infection, cells were infected with virus (MOI 1.5) for 1 hour at 37°C. The inocula were removed, rinsed with PBS, and cells were further incubated up to 7hpi at 37°C. Cells were then stained with anti-dsRNA as described above.”

And in the legend of Figure 4—figure supplement 1:

“Figure 4—figure supplement 1. Localization of viral capsids and double-stranded RNA at respectively 0.5 hpi and 7 hpi in FACS-sorted GFP-positive Vero cells transiently expressing Rab5a-eGFP WT or CA and infected with the two viral variants. (A) Vero cells infected with MP4 and MP4-97R/167G were fixed at 0.5 hpi. Viral capsid localization was imaged with an anti-VP2 Ab (in red). Magnified areas are highlighted in white box and displayed at left bottom of merged image. Scale bar: 20 µm. (B) Vero cells infected with MP4 and MP4-97R/167G were fixed at 7 hpi and stained with anti-dsRNA J2 Ab to highlight viral replication. The images shown are representative examples of those used for the quantifications presented in Figure 4D. Scale bar: 400 µm.”

Reviewer #3 (Recommendations for the authors):The manuscript should be enhanced by additional biophysical and computational studies to establish the link between the link between increase in net positive charge and stability.

We have performed additional electrostatic calculations with APBS as suggested by the reviewer and their analysis corroborates the impact of the mutations on the capsid surface. The results are presented as a new figure (Figure 5—figure supplement 1) and discussed in the text (Line 208).

Figure 5—figure supplement 1. Electrostatic surface potential calculated by Adaptive Poisson–Boltzmann Solver (APBS) from −5 kT/e (red) to +5 kT/e (blue).

Result section page 6, line 208: We analyzed the impact of VP1-97R and VP1-167G mutations on their respective local environments at pH 5 and pH 7 with the Adaptive Poisson-Boltzmann Solver (APBS)^33,34^. As displayed in Figure 5—figure supplement 1, both L97R and E167G mutations are inducing electrostatic changes at the surface of the capsid within the region of interest and the changes are particularly significant at pH 5. Furthermore, analysis with DynaMut server^35^ reveals that the two mutations affect interaction networks (Figure 5—figure supplement 2). The VP1-L97R mutation is predicted to reduce hydrophobic interactions between amino acid 97 and its neighbors VP1-245Y and VP1-246P while the VP1-E167G mutation causes a loss in hydrogen bonding capacity to VP1-165S and reduces the net negative charge. This is consistent with analyses of vibrational entropy change (Figure 5—figure supplement 3), indicating that the presence of two mutations results in enhanced local dynamics, which has previously been correlated with reduced capsid stability^36,37^. Consistently, the predictions of Gibbs free energy change (ΔΔG) induced by these mutations further support that both mutations induce destabilization of the capsid structure, regardless of pH and temperature (Figure 5-table supplement 1).

We also attempted to improve our knowledge on the exact effect of the mutations on the virus capsid using molecular dynamics (MD) simulations at different pHs and different temperatures. However, the large size of the full capsid has forced us to extract only a subpart of it (a capsid pentamer), in order to set up a system realistically treatable (~468 K atoms for the solvated system). This approach suffers from the fact that we do not model the full capsid, and thus some parts of the capsid lack the proper molecular environment. The truncated model of the capsid highlighted, within the limited timescale sampled, no indications of significant changes due to temperature, pH and mutational effects (Author response image 1). Thus, we can conclude that if a correlation exists between these conditions and capsid stability, this happens on longer timescales and/or depends on additional factors still poorly understood.

**Author response image 1. sa2fig1:** Molecular dynamic simulations of a subpart (corresponding to a capsid pentamer) of the wild type VP1 capsid (a) and the L97R-E167G mutated version (b), each at pH 5 and 7 and at 37°C (310K) and 50°C (323K). The extraction of the subpart and the lack of physical constraints around the external-facing regions are inducing artificial moves during the simulation, and preventing its convergence, as witnessed by the high variability of root mean square deviations (RMSDs).

Methodology: We set up simulations at different pH (5 and 7) and at different temperatures (37°C and 50°C), each during 200ns using GROMACS (Bekker *et al.*, Physics computing, 1993; Abraham *et al.*, SoftwareX, 2015). Due to the large size of the viral capsid, only a subpart of it was simulated. A manual cut around the region of interest was performed, reducing the size of the studied system while giving some sufficient margin to observe the mutations sites, resulting in a complex with 49’905 atoms (468’455 atoms after generating the solvation box).

We set up the system with CHARMM force field and used TIP3P water as solvent. We performed one minimization step for 20 ps, with an energy tolerance of 1000.0 kJ/mol/nm. We employed the Verlet cut-off scheme, treated long-range electrostatics using the Particle Mesh Ewald (PME) method, and applied constraints on all bonds involving hydrogen atoms using the LINCS algorithm. Following minimization, we performed the equilibration in five steps. The first two steps were performed for 1 ns each, using a 1 fs time step. The subsequent three steps utilized a 2 fs time step for 500 ps each. Throughout the equilibration, we maintained position restraints on the protein. We used the Nose-Hoover thermostat for temperature coupling, with the system gradually heated to the target temperature. From the second step onward, we introduced pressure coupling using the Parrinello-Rahman barostat, set to 1.0 bar with isotropic pressure coupling. We still employed the Verlet cut-off scheme for non-bonded interactions and the Particle Mesh Ewald method for long-range electrostatics and applied the LINCS algorithm to constrain hydrogen-containing bonds. After equilibration, we performed the production MD simulation for 200 ns using a 2 fs time step. The system temperature was maintained at the target temperature using the Nose-Hoover thermostat, while pressure was kept at 1.0 bar using the Parrinello-Rahman barostat with isotropic coupling. We continued to use the Verlet cut-off scheme, the Particle Mesh Ewald method and the LINCS algorithm. Trajectory data, including coordinates, velocities, and forces, were saved every 100 ps for subsequent analysis. Results were compiled and analyzed using the Python package MDAnalysis ([Gowers *et al.*, Sci Py, 2016; Michaud-Agrawal *et al.*, J. Comput. Chem. 2011]). The structural cut performed to make the system simulable had a strong impact on the overall stability, inducing important artificial moves throughout the entire system.

Also, any distinction between the entry pathway of variants needs to be considered.The entry pathway of the L97R/E167G variant, in light of enhanced infectivity under elevated endosomal pH, requires more probing. How would close to neutral pH protect uncoated particles?Are the authors convinced that MP4 and the L97R/E167G variant show the exact same pattern for cellular surface association? Has SCRAB2 association *in vitro* or cellular association been measured/quantified by any biophysical method? Also, is it possible that the variants enter cells through some other pathway – macropinocytosis, for example?

Our experiments demonstrate that both variants localize to early endosomes within 30 minutes (Figure 4—figure supplement 1A) and that afterwards, the MP4 variant enters cells via pH-dependent endocytosis while MP4 97R/167G releases its genome before endosomal acidification (Figure 4C).

The MP4 97R/167G variant is stable at pH 7 but becomes unstable at acidic pH or upon SCARB2 binding, unlike MP4 which remains stable under these conditions. This suggests that the 97R/167G variant’s reduced capsid stability needs a single trigger (acidic pH or SCARB2 binding in the endosome) to start uncoating. However, we do not think that there is a risk for MP4 97R/167G to uncoat at neutral pH outside the cell. Indeed, several recent studies have highlighted that SCARB2 is localized primarily in lysosome membrane (Kobayashi *et al.*, 2020; Kuronita *et al.*, 2002). Furthermore, EV-A71 does not seem to bind this protein to attach to host cells (Nishimura *et al.*, 2024; Guo *et al.*, 2022). We also observed this with a human EV-A71 variant and a derivative with the same two mutations (Tseligka *et al.*, 2018; Weng *et al.,* 2023), as these two variants bind as efficiently wild-type or SCARB2 CRISPR-Cas9 knock-out cells (Author response image 2). Therefore, the unstable MP4 97R/167G variant will not bind SCARB2 before endocytosis, and uncoating should not start before the virus reaches endosomes, where SCARB2 and viral particles come in close proximity.

Finally, in cells treated with heparinase to deplete HS, or in Caco-2 cells known to have low HS levels, both variants exhibit similar binding in presence or absence of SCARB2 (Author response image 2). In addition, as shown in the manuscript, MP4-97R/167G demonstrates enhanced cell binding due to its higher affinity for HS (manuscript Figure 2A and Author response image 2), but HCQ does not impact virus binding, and HCQ effect on viral replication persists even after HS removal or inhibition of cell sulfation by sodium chlorate treatment (manuscript Figure 1C and Figure 2A). Altogether these findings rule out the hypothesis that the differential sensitivity of the two variants to lysosomotropic drugs is related to different interactions with SCARB2 or HS at the cell surface.

To conclude, our data demonstrate that mutations involved in conferring affinity for HS can in parallel and independently decrease capsid stability and favor pH independent uncoating. However, we did not investigate further the precise entry route for MP4-97R/167G, as we believe that this is beyond the scope of the current study.

**Author response image 2. sa2fig2:** Binding of (a) EV71-VP1-97R167G and (b) EV71-VP-97L167E in RD and Caco-2 cells with and without SCARB2 (Cas9 CRISPR KO using two different guide RNAs, g1 and g2) and/or HS expression (heparinase treatment). Viral loads were measured by RT-qPCR in whole cell extracts and are expressed as the mean ± SEM.

The MP4 particle and the variants can be subjected to Differential Scanning Calorimetry (DSC) to understand the difference in stability and for semi-quantification of the difference among variants. Does progressive increase in overall positive charge further decrease the stability of variants? Likewise, the in silico stability analysis should be expanded to analyze subunit interaction.

As recommended, we performed particle stability thermal release assay (PaSTRy assay) to compare the susceptibility of the two variants to thermal denaturation at different pHs. As shown in Author response image 3, and as expected, MP4 exhibits comparable thermal stability after incubation at both pHs. In contrast, MP4-97R/167G shows reduced stability at lower pH. However, the results are difficult to interpret due to the odd shape of the curves for the double mutant. These varying peak depths could stem from intrinsic characteristics of the mutant virus (presence of mixed states of the particles throughout the assay due to lower stability). For this reason, we decided not to include these results in the manuscript.

**Author response image 3. sa2fig3:** PaSTRy assay was performed by incubating virus in citrate phosphate buffer adjusted to pH7 and pH5. Viral RNA release from capsid was detected using SYBR green II dye when the virus capsid was heating gradually with temperature increase of 1°C from 25°C to 95 °C.

Instead, we performed additional *in silico* stability analysis using several methods. A computational analysis of electrostatic potential using APBS has been added in the manuscript, which corroborates the experimental observations (Figure 5—figure supplement 1). We also performed MD simulations, at different temperatures and different pH. However, the limitations of the MD setup that we discussed above are making its result not conclusive, and we are not able to discuss subunit interactions from the trajectories we obtained.

The effect of both receptor binding and low pH on the morphology of the particles *in vitro* should be tested. The MP4 particles do not seem to form A or B particles even at pH 5.0, will this happen with a conjunction of receptor binding and low pH?

As EV-A71 is known to bind poorly to SCARB2 at the cell surface (see response to previous comment), it is thus difficult to compare the effect of SCARB2 binding at neutral and acidic pH. We would like to note that, as indicated in the literature, incubation at acidic pH is necessary for performing cryoEM of the EV-A71/SCARB2 complex. Previous studies have also required the use of a mutated EV-A71 strain and the addition of an expansion inhibitor (Zhou *et al.*, 2019). This suggests that the interaction with SCARB2 is very unstable in standard conditions and also that the virus may quickly detach after binding. Nevertheless, to address the reviewer comment, we conducted nsEM of MP4 incubated with soluble SCARB2 for 30 min at neutral or acidic pH. We could not find evidence that SCARB2 and MP4 form stable complexes. Based on the analysis of 2D classes (Author response image 4), particles appear smooth and free of any density on the surface. However, we did observe an increase in the number of empty capsids following the incubation with SCARB2 at pH 5 compared to pH 7.5 (Author response image 4). These data suggest that other triggers occur in the lysosomal compartment to promote genome release in the host cytoplasm. We included the data obtained at pH 7.5 in Figure 5—figure supplement 4 as part of the discussion on the differential effects of SCARB2 binding on different mutants.

**Author response image 4. sa2fig4:** Raw micrographs and 2D classes of MP4 incubated with soluble SCARB2 for 30 minutes at neutral or acidic pH. Red squares indicate 2D class averages with strong staining in the middle of the particle indicating partially open capsids with low density of protein and genome components, which is consistent with empty capsids.

The capsid alterations in the reconstructions require adequate explanation. The images of the variant at low pH indicate asymmetric conformational change – under such circumstances, application of icosahedral symmetry may cause misleading density interpretation. The 5-fold axes appears to be altered in the variant at low pH – can this be explained in the context of uncoating-related alterations?

We fully agree with the reviewer, in that the use of icosahedral symmetry is not appropriate for viral capsids that exhibit this expanded asymmetric phenotype. To address the reviewer comment, we have performed Ab initio reconstruction of these particles without imposing symmetry (i.e., C1). The resulting 3D map still shows an expanded particle compared to the reconstruction obtained at neutral pH, which is consistent with expansion observed in raw images and 2D class averages. Icosahedral 3D map in the original figure has been replaced with an asymmetric reconstruction in the revised manuscript (Figure 5B).

Do the mutations correlate with calcium ion coordination, subunit interfaces or pocket factor binding in the context of the capsid structure? Do these residues influence the conformation of the dynamic regions of the capsid – the VP4 peptide or the N-terminus of VP1 – needed for endosomal membrane rupture?

We agree with the reviewer that extensive simulations performed in the indicated conditions could enhance our understanding of the effect of the mutations on conformation of the capsid structure. However, as discussed above, MD simulations present significant limitations when studying such large systems and force fields models for divalent cations are usually not very reliable. For this reason, we have not added calcium ions to our model system but only NaCl (see Author response image 1 and associated methods).

Reviewer #4 (Recommendations for the authors):A few specific comments/changes that will improve the manuscript are provided below:1. Taxonomically, it should be written as Enterovirus A71. The hyphen (-) in line 46 between Enterovirus and A71 must be removed.

We have implemented this change consistently throughout the manuscript.

2. A few references are missing in the introduction or some statements may be based on earlier or later references but this is not immediately evident (e.g. line 45 and line 47).

The missing references have been added to the manuscript.

3. Please adjust line 52 and others where SCARB2 is suggested to be important for internalization. It can be referred to as an entry receptor as uncoating is demonstrably an entry step. Alternatively, refer to it as the uncoating receptor.

We have introduced this change.

4. Lines 59-63 are a summary of another study but given the lack of contextualization to the human data this should be removed and/or the human data should be better incorporated in the introduction.

Additional references have been included to provide better context in relation to the human data.

5. Please add the quantification of Figure 1B.

We have added the quantification data; however, for clarity and because these are control experiments, we have moved them to the Figure 1—figure supplement 1C.

6. Line 153 states that MP4 uncoating was completely blocked but this is not true from Figure 3B. Please adjust.

We have revised the text according to the reviewer comment. The sentence "completely blocked" has been changed to "greatly inhibited" on page 5, line 164 to accurately reflect the data presented in Figure 3B.

7. The MOI used for the different experiments is not clear. Please mention this in the figure captions.

We have added the MOI used for the different experiments as requested.

8. Please perform statistical analysis on Figure 2F.

We have included the statistical analysis accordingly and added the P value on the figure.

9. Please provide the individual panels for some of the fluorescence images such as 1B and 3C.

We have included the individual panels for these fluorescent images (Figure 1—figure supplement 1B and revised Figure 3C, respectively) as requested.

10. Please provide a set of representative images corresponding to Figure 4D.

We have included the representative images for this figure in the Figure 4—figure supplement 1B.

11. In the discussion, it is stated that existing literature aligns with the findings. However, this is existing literature on mice but not humans which is in contrast. Please adjust this section based on the comments above.

As suggested, we have modified this sentence to clarify, and we now provide more details on the differences between observations in animal models and human in the conclusion paragraph (page 9, lines 332-336 and 364-376):

“These studies, which report a correlation between capsid instability, earlier uncoating, and attenuation in mouse models aligns with our findings. This suggests that amino acid changes affecting capsid stability can significantly impact various aspects of the virion and its life cycle, including sensitivity to environmental factors, receptor interactions, and infection rates.”

“Of note, although strong HS-binder are clearly attenuated in mice, the situation in humans is more puzzling as strains with Q or G at position VP1-145 have been associated with severe neurological cases and the methodology used in those studies excluded emergence of mutation during cell culture^19,20^. Experimental work in human relevant tissue culture models also suggest that binding to HS provides an advantage to the virus^22-24^. Furthermore, we previously showed that intermediate HS-binding affinity can result in increased virulence even in mouse models^28^. We are currently investigating if there is a lack of trapping and/or limited impact on viral stability under these conditions. ”

12. In line 330, it is stated that the pH of the gastrointestinal tract is acidic. However, this is not true for the entirety of the GI tract and some parts are close to neutral pH.

We have amended the text by changing "gastrointestinal tract" to "stomach" on page 10, line 360.